# Topologically associating domains are ancient features that coincide with Metazoan clusters of extreme noncoding conservation

Nathan Harmston [1,2,3], Elizabeth Ing-Simmons [1,2,4], Ge Tan[1,2], Malcolm Perry[1,2], Matthias Merkenschlager[2,4] & Boris Lenhard [1,2,5]

Developmental genes in metazoan genomes are surrounded by dense clusters of conserved noncoding elements (CNEs). CNEs exhibit unexplained extreme levels of sequence conservation, with many acting as developmental long-range enhancers. Clusters of CNEs define the span of regulatory inputs for many important developmental regulators and have been described previously as genomic regulatory blocks (GRBs). Their function and distribution around important regulatory genes raises the question of how they relate to 3D conformation of these loci. Here, we show that clusters of CNEs strongly coincide with topological organisation, predicting the boundaries of hundreds of topologically associating domains (TADs) in human and *Drosophila*. The set of TADs that are associated with high levels of noncoding conservation exhibit distinct properties compared to TADs devoid of extreme noncoding conservation. The close correspondence between extreme noncoding conservation and TADs suggests that these TADs are ancient, revealing a regulatory architecture conserved over hundreds of millions of years.

[1] Computational Regulatory Genomics, MRC London Institute of Medical Sciences, London W12 0NN, UK. [2] Institute of Clinical Sciences, Faculty of Medicine, Imperial College London, London W12 0NN, UK. [3] Program in Cardiovascular and Metabolic Disease, Duke-NUS Graduate Medical School, 8 College Road, Singapore 169857, Singapore. [4] Lymphocyte Development, MRC London Institute of Medical Sciences, London W12 0NN, UK. [5] Sars International Centre for Marine Molecular Biology, University of Bergen, N-5008 Bergen, Norway. Correspondence and requests for materials should be addressed to N.H. (email: nathan.harmston@duke-nus.edu.sg) or to B.L. (email: b.lenhard@imperial.ac.uk)

In Metazoa, many genes involved in developmental regulation are surrounded by syntenic arrays of conserved noncoding elements (CNEs)[1–3]. These CNEs exhibit extreme levels of conservation over many base pairs, and in some cases more than the equivalent conservation of protein-coding regions[4]. Several studies have shown that individual CNEs can act as transcriptional enhancers to drive complex spatiotemporal expression patterns[5, 6]. However, no known source of selective pressure is able to account for their extreme conservation[7].

In enhancer assays, CNEs drive reporter expression which corresponds to the expression patterns of nearby developmental regulators, suggesting that they act as regulatory elements for these genes[8–10]. Different CNEs drive expression in different spatiotemporal domains[5, 6, 11], resulting in complex expression patterns driven by combinations of regulatory elements. The maintenance of synteny between CNEs and their target genes over large evolutionary distances results from the necessity of keeping regulatory elements in *cis* with the gene under long-range regulation[12]. We should emphasise that in regulatory genomics conservation of synteny corresponds to the conservation of collinear arrangement of genes and other conserved sequences between the genomes of two species. This syntenic organisation of clusters of CNEs around key developmental genes, called genomic regulatory blocks (GRBs)[3, 13], supports the idea that they are ensembles of regulatory elements that are involved in regulating these genes (Supplementary Fig. 1a). In addition to this target gene (or genes in the case of gene clusters) under developmental regulation, a GRB can harbour several other genes that are not detectably regulated by these elements (bystander genes). Target and bystander genes differ with respect to their promoter structure, patterns of epigenetic modification and range of biological functions[3, 14, 15].

Long-range regulation depends on the interaction of a target gene promoter with enhancers that can be located up to a megabase away, which need to be brought into close physical proximity in the nucleus. Insights into interactions at this scale and their roles in development and differentiation have been provided by the development of chromatin conformation capture methods[16]. Interactions between regulatory elements located within the introns of bystander genes and the promoters of target genes have been identified[17]. The 3D structure of vertebrate Iroquois clusters, which are known GRBs, is highly conserved across vertebrates[18], and is thought to result from enhancer sharing and co-regulation of members of the cluster during development. Hi-C has revealed regions of the genome which preferentially self-interact, known as topologically associating domains (TADs) or contact domains[19, 20]. Regulatory elements and genes preferentially interact within the same TAD, suggesting that the boundaries of TADs may act to restrict the influence of enhancers[21, 22].

Despite containing cell type-specific promoter–enhancer interactions, the boundaries of TADs in mammals is largely invariant across different cell types[19, 20, 23, 24], and between species[19, 25]. TADs have previously been found to correspond with other large-scale genomic features, including replication domains[26, 27] and Polycomb-repressed domains[28], and to have boundaries that coincide with conserved CTCF binding[25]. While it appears that human, mouse and *Drosophila* chromosomes are segmented into TADs along their entire length[19, 28], in *Caenorhabditis elegans* their occurrence varies both between and along different chromosomes[29]. Plant chromosomes do not seem to be organised into TADs, except for isolated TAD-like structures at a limited set of loci[30]. At longer length scales, Hi-C data suggest that TADs are organised into two major compartments, which correspond to open (compartment A) or closed chromatin (compartment B), which tend to self-associate within the nucleus[31]. TADs may switch compartment depending on the activity of genes within them[23], and therefore it has been suggested that TADs form the '*regulatory units*' of the genome[32].

The stability of TADs, the variation in their size and their ubiquitous presence across different metazoan phyla lead us to question how these domains relate to the regulatory domains of developmental genes. Given that our previous work strongly suggested that GRBs correspond to this type of domain[3, 5, 13], and that little is known about their relationship with genome-wide 3D organisation, here we investigate the relationship between GRBs and TADs in both vertebrates and invertebrates. We find a strong concordance between the extent of a subset of TADs and clusters of CNEs in both humans and *Drosophila*. This indicates that these TADs, which have a distinct set of features compared to those lacking CNEs, correspond to the regulatory domain of the gene under long-range regulation, potentially representing a functionally distinct class of TADs. The presence of this relationship in different Metazoan phyla indicates that this type of organisation has been under intense selective pressure, and acting on the same functional subset of genes, at least back to the common ancestor of chordates and arthropods.

## Results

In this paper, we define a CNE as a noncoding element that has a high percentage identity over a defined number of base pairs between two species (see Supplementary Methods). We operationally define GRBs as discrete regions of the genome with a high density of syntenic CNEs. The locations of putative GRBs and their approximate spans can be visualised by plotting the density of CNEs in a sliding window (Fig. 1a); this visualisation is available for multiple genomes in the Ancora browser[33].

We developed a CNE clustering approach that robustly estimates the extent of GRBs, based solely on the distribution of syntenic CNEs in the genome (see Supplementary Methods). The procedure is shown schematically in Supplementary Fig. 1b. This allowed the examination of the physical extent and boundaries of putative GRBs, and their comparison with higher-order chromatin organisation.

Hi-C data sets were obtained[19, 23, 28] and processed to generate genome-wide interaction maps in different species and cell lineages (see Supplementary Methods). We used Hi-C *directionality index* as a visual indicator of topological organisation. Directionality index quantifies the degree of upstream or downstream interaction bias, which was then processed to segment the genome into a discrete set of TADs for downstream analysis[19, 34].

**Identification of GRBs**. We applied our CNE clustering method to generate sets of GRBs between species at various evolutionary distances. GRBs were identified in the human genome using the distribution of CNEs observed in comparisons with opossum (160 Mya separation), chicken (320 Mya), and spotted gar[35] (430 Mya). GRBs between human and chicken (hg19-galGal4) were identified using a number of thresholds: 819 GRBs were identified using CNEs showing 70% identity over 50 bp (Fig. 1b), 672 using 80% over 50 bp and 468 using 90% over 50 bp. Regardless of the threshold used to identify CNEs between humans and chicken, the density of CNEs and associated GRB predictions were highly concordant (Fig. 1 and Supplementary Fig. 2). A total of 1,160 GRBs were predicted between humans and opossum (hg19-monDom5) using CNEs showing 80% identity over 50 bp and 719 between human and spotted gar (hg19-lepOcu1) using CNEs showing 70% identity over 30 bp (Supplementary Table 1).

As evolutionarily conserved features, we expect the generated GRBs to have stable boundaries. Indeed, the span of GRBs

identified using hg19-galGal4 show close agreement with the distribution of CNEs identified from hg19-monDom5 and hg19-lepOcu1 comparisons (Fig. 1c). Examination of several

GRBs revealed a marked correspondence of their boundaries between evolutionarily distant species. Comparison of the boundaries of hg19-galGal4 GRBs that overlapped with

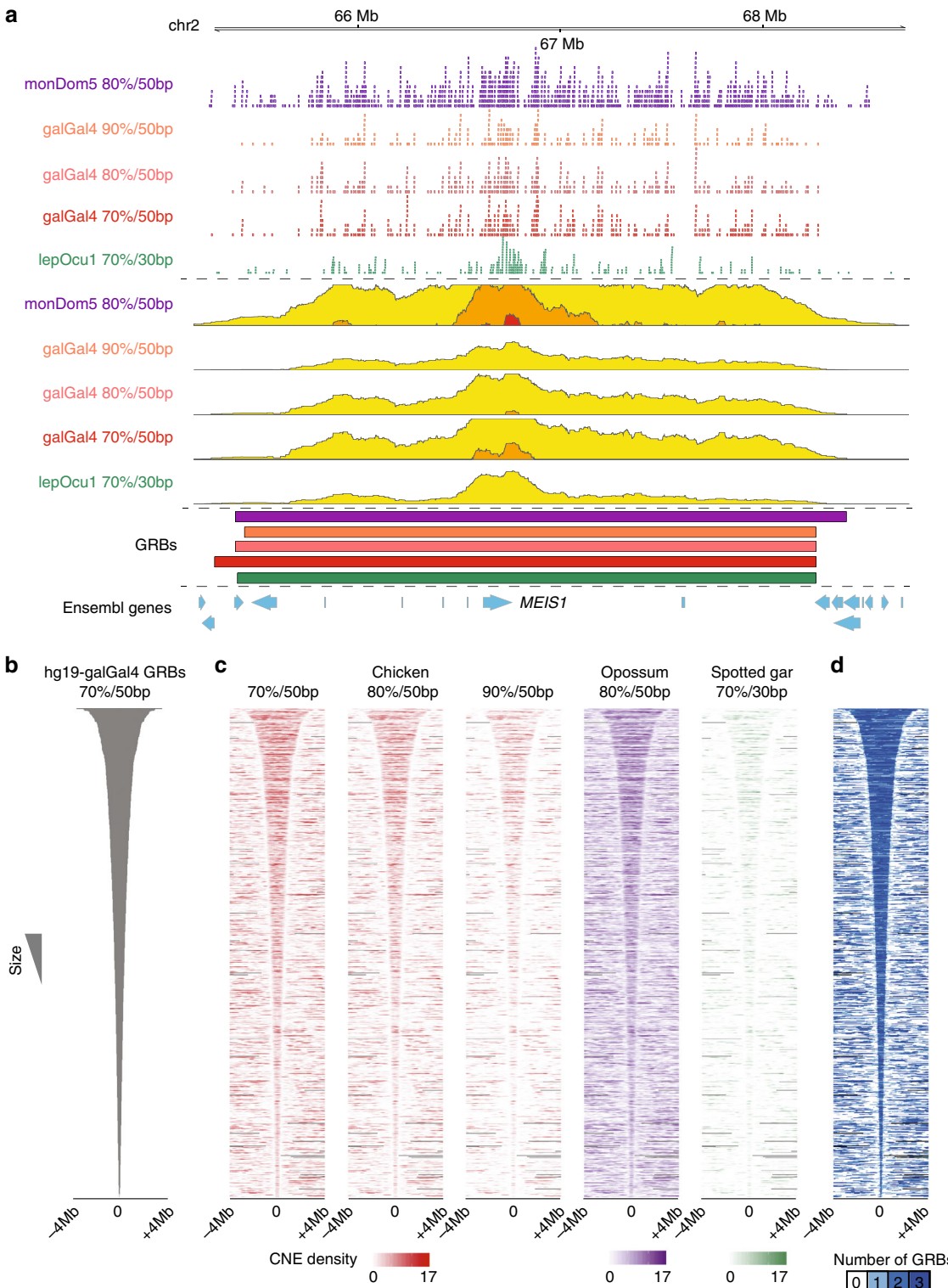

**Fig. 1** The boundaries of GRBs are highly consistent regardless of the thresholds or species involved. **a** The human *MEIS1* locus is spanned by arrays of conserved noncoding elements (CNEs) identified in comparisons with opossum, chicken and spotted gar. These CNEs can be visualised as a smoothed density, shown here as a *horizon plot*. The boundaries of the proposed GRBs at the *MEIS1* locus are highly consistent regardless of the species or thresholds involved. **b** All hg19-galGal4 GRBs centred and ordered by length of the GRB. **c** Distribution of CNEs in a window of 8 Mb around the centre of the hg19-galGal4 GRBs for different sets of CNEs. **d** Overlap of putative GRBs obtained using comparisons between hg19-monDom5 and hg19-lepOcu1 with these sets of GRBs identified using CNEs using hg19-galGal4 (70%/50 bp)

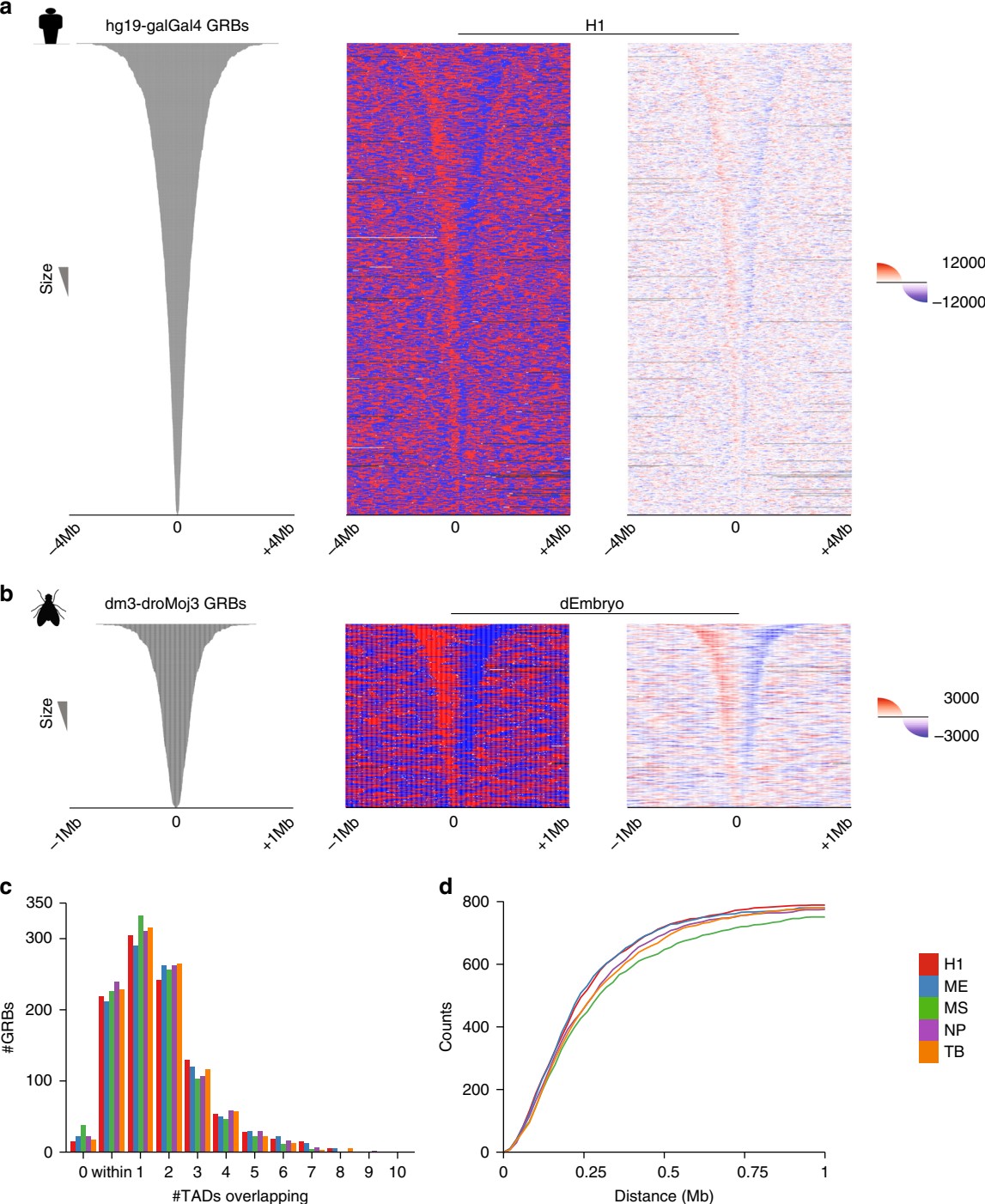

**Fig. 2** The boundaries of GRBs predict the boundaries of TADs in multiple evolutionarily distant species. **a** Heatmaps representing H1-ESC directionality index spanning an 8 Mb window around the centre of putative hg19-galGal4 GRBs. Showing both the overall direction (*middle panel*, *red* for downstream, *blue* for upstream) and the average raw directionality score in 5 kb bins (*right panel*). **b** Heatmaps of *Drosophila* embryo Hi-C directionality index spanning a 2 Mb window around the centre of dm3-droMoj3 GRBs. Showing both the overall direction (*middle panel*, *red* for downstream, *blue* for upstream) and the average raw directionality score in 1 kb bins (*right panel*). **c** A large number of GRBs were found to be located within individual TADs (identified using HOMER) or overlapping only a single TAD, regardless of cell lineage (H1-ESC (H1), mesenchymal stem cells (MS), mesendoderm (ME), neural progenitor cells (NP) and trophoblast-like (TB)). **d** Cumulative distribution of distance to nearest TAD (HOMER) boundaries from GRB boundaries in different cell lineages considering both edges, i.e., both the start and end positions of a GRB lie within *X* kb of the nearest TAD start and end

individual hg19-monDom5 GRBs found that 276 (50%, 554 in total) had boundaries that differed by less than 150 kb. A subset of hg19-galGal4 GRBs appear to have similar boundaries to hg19-monDom5 and hg19-lepOcu1 GRBs (Fig. 1d), including GRBs containing well-known developmental regulators (Fig. 1a

and Supplementary Fig. 2). This suggests that the span of several GRBs is invariant to the evolutionary distances involved in identifying them.

Two main problems can occur when predicting GRB spans from CNE density. If the synteny between two adjacent GRBs is

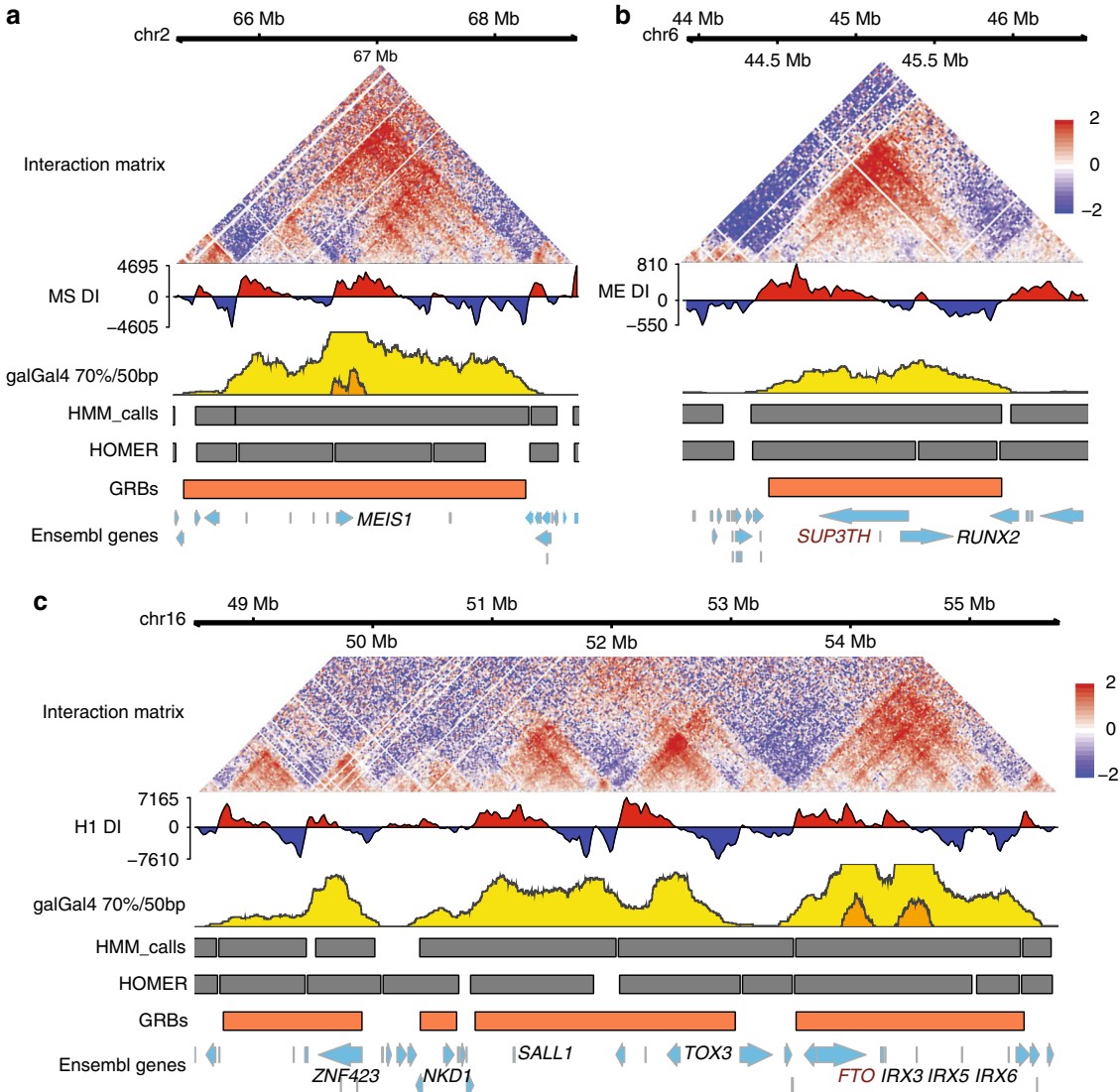

**Fig. 3** Examples of genomic regulatory blocks and their associated interaction landscapes in human. GRBs at several human loci show strong association with the structure of regulatory domains proposed from Hi-C. **a** The GRB containing *MEIS1* (chr2:65270920-68723490) accurately predicts the span of regulatory interactions defined by Hi-C. **b** The region located at chr6-44198640-46071520 contains both the transcription factor *RUNX2* and its bystander gene *SUP3TH* (shown in *brown*), both of which are located within a GRB which predicts the topological organisation of the locus. **c** A region located on chr16:48476700-55776880 in hg19 contains several GRBs containing important developmental regulators, including *IRX3/5/6*, *TOX3*, *SALL1*, *NKD1* and *ZNF423*, which exhibit strong concordance with TADs. The *IRX3/5/6* locus contains homeobox proteins which have multiple functions during animal development and contains a well-known bystander gene *FTO* (shown in *brown*), which contains an intronic enhancer which drives expression of *IRX3*[38-41]

conserved in the species used for comparison and the GRBs are extremely close together, there is no information from the CNE density alone that would enable their separation (Supplementary Fig. 2a, d). For that reason, we expect that a fraction of estimated spans will contain multiple adjacent GRBs, and predict that this will be more prevalent with the longest span predictions (i.e., top set of predictions in Fig. 1b). Alternatively, since the density of CNEs along a GRB is non-homogeneous, a single GRB can be split into two or more putative regions. We expect this will be more prevalent in predictions of smaller GRBs (i.e., lower set of predictions in Fig. 1b).

The identification of GRBs and the concordance of their boundaries over multiple species and thresholds suggests that our clustering method is robust and that GRBs represent evolutionarily conserved structures. For our human-centric comparisons we used GRBs identified using CNEs conserved between humans and chicken at 70% identity over 50 bp, which

provide the best compromise between GRB coverage and separation of adjacent GRBs.

**GRB boundaries coincide with the boundaries of TADs.** Visualisation of Hi-C directionality index around predicted GRB spans revealed a striking pattern of abrupt transitions in directionality index at the boundaries of GRBs in both humans (Fig. 2a) and *Drosophila melanogaster* (Fig. 2b), indicative of TAD boundaries (see Supplementary Methods). The funnel shape of the Hi-C directionality index indicates that TAD boundaries closely trace our estimated GRB boundaries. The concordant pattern of TAD and GRB boundaries revealed in Fig. 2a, b is compelling, suggesting a functional relationship between these two sets of independently identified features. As expected, the largest of the predicted GRB spans show multiple directionality changes within the region, in line with our expectation that these

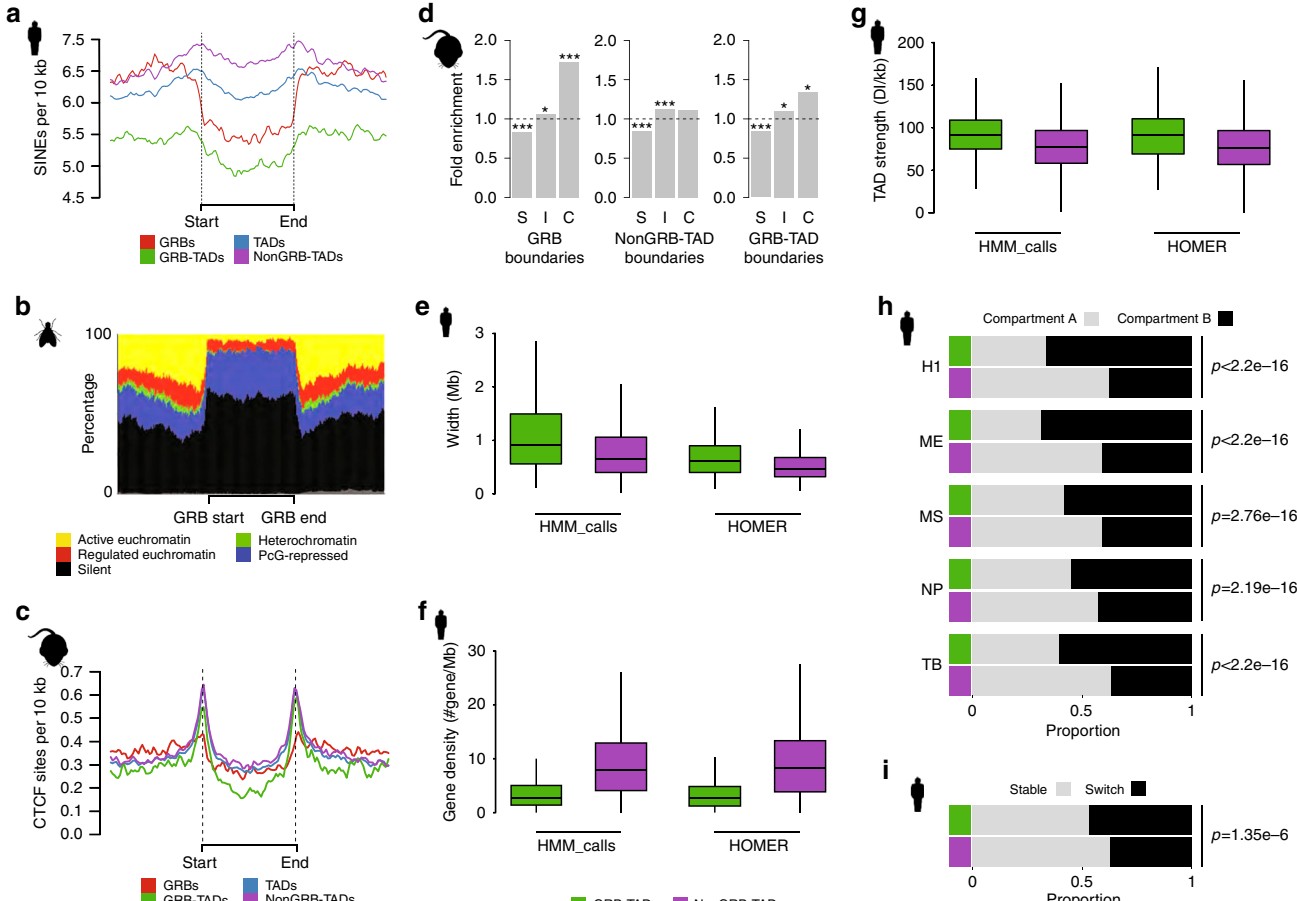

**Fig. 4** Several sets of features distinguish between TADs associated with extreme conservation from those without. **a** Depletion of SINE elements within GRB-TADs compared to non-GRB-TADs (using H1 HOMER TADs) reflects the selective constraint on these regions against the retention of repeat element insertion. **b** GRBs in *Drosophila* Kc167 cells are mainly associated with inactive (*black*) and Polycomb-repressed chromatin (*blue*) and represent functionally coherent regions. Active and regulated chromatin correspond to different types of euchromatin. There appears to be a change in the proportion of constitutively active chromatin (*yellow*) at the boundary regions identified as GRBs. **c** CTCF sites are depleted within GRBs. CTCF sites per 10 kb plotted across GRBs and flanking regions of equivalent size to the GRBs normalised to show signal relative to GRB boundaries and Loess-smoothed. **d** Enrichment for different patterns of CTCF binding at different genomic features (*S*pecific, *I*ntermediate, *C*onstitutive). Constitutive CTCF peaks are enriched within 10 kb of both GRBs and GRB-TAD boundaries (binomial test *p*-values: *\*p < 0.05, \*\*p < 0.01, \*\*\*p < 0.001*). **e** Distribution of TAD width reveals GRB-TADs are significantly longer than non-GRB-TADs identified in human H1 cells using either HOMER (median width 620 kb vs. 460 kbp, *p < 1e−6*) or HMM_calls (median width 920 vs. 680 kb, *p < 1e−6*). **f** Human H1 GRB-TADs are associated with lower protein-coding gene density than non-GRB-TADs identified using HOMER (median no. genes 2.63 vs. 8.33 *p < 1e−6*) or HMM_calls (median no. of genes 2.65 vs. 8.33 *p < 1e−6*). **g** GRB-TADs are significantly stronger than non-GRB-TADs identified in human H1 cells using HOMER (median strength 90.88 vs. 75.81, *p < 1e−6*) or HMM_calls (median strength 91.77 vs. 77.68, *p < 1e−6*). **h** GRB-TADs (HOMER) are preferentially associated with compartment B in all of the lineages investigated. **i** GRB-TADs are more likely to switch compartment in at least one of the five lineages investigated (i.e., A–B or B–A) than non-GRB-TADs

may represent multiple closely spaced GRBs. The association between GRBs and directionality index is stable regardless of the cell line used (Supplementary Fig. 4a). To further explore this relationship, we compared the span of our sets of GRBs with TADs identified in several human cell lineages[23] and *Drosophila* embryos[28]. Regardless of either the cell lineage or the method used to identify TADs, a large number of GRBs were found to be located within individual TADs or overlapping a single TAD (Fig. 2c and Supplementary Fig. 3a). Investigating the distances between putative GRB boundaries and the nearest TAD boundary revealed a close association (Fig. 2d, Supplementary Fig. 3b, e, g and Supplementary Table 2), with both edges of 235 GRBs (29%) lying within 120 kb of the nearest TAD boundary in H1-ESCs (*p < 1e−5*). The relationship between CNE density and topological structure was also observed in high-resolution Hi-C[20] (Supplementary Fig. 5). We confirmed this association in an evolutionarily distant species using a set of GRBs identified between

*D. melanogaster* and *Drosophila mojavensis* (dm3-droMoj3) (63 Mya). In all, 317 GRBs were identified using CNEs showing 96% identity over 50 bp. Again, the majority of GRBs were located within or overlapped one TAD (Supplementary Fig. 3c, d), with a close association between TAD boundaries and GRB boundaries (Supplementary Fig. 3f). Therefore, there is a strong genome-wide correspondence between the genomic regions enriched for extreme noncoding conservation and those identified as TADs. This relationship is robust with respect to cell lineage, species and the computational method used to identify TADs.

We tested whether the correspondence between the TADs and GRBs could be a consequence of clusters of enhancers being present in these regions. The spans of neither GRBs nor TADs coincide with the distribution of H3K27ac or H3K4me1 (Supplementary Fig. 4b, c respectively). This observation, along with the lack of predictive power in using H3K27ac (and other

histone modifications) to predict the extent of TADs[36], implies that the potential enhancer activity of CNEs is not sufficient to explain the observed pattern.

As predicted by our genome-wide analysis, individual loci of known target genes showed a strong concordance between CNE density and topological structure (Fig. 3 and Supplementary Fig. 6). The *MEIS1* GRB encompasses all of the CNEs identified as enhancers in reporter assays[37], with a strong resemblance between GRB boundaries, which are highly concordant regardless of the species used to identify them (Fig. 1a), and the topological organisation of this locus as defined by Hi-C (Fig. 3a). A similar pattern is observed at the locus containing *RUNX2* (Fig. 3b). The boundaries of the IRX3/5/6 GRB are consistent regardless of the species used to identify CNEs (Supplementary Fig. 2a), and highly predictive of TAD boundaries, showing a strong concordance with both the directionality index and interaction matrix (Fig. 3c). Importantly, the TAD contains the *FTO* gene, whose intronic elements are involved in the regulation of *IRX* genes[38–41]. Our method identified the region containing *TOX3* and *SALL1* as a single GRB (Fig. 3c); however, the boundary between the regulatory domains of these genes identified in enhancer screens[9, 42] is reflected by a TAD boundary located in the middle of this region. The regulatory landscape of the HoxD cluster has previously been found to be better predicted by synteny than by its topological structure[43], and indeed at this locus the span of the potential interactions closely resembles the distribution of CNEs (Supplementary Figs 6a, 7a).

While CNEs cluster around orthologous transcription factors in both vertebrates and arthropods, CNEs themselves exhibit little sequence similarity between phyla[3, 44]. *Drosophila* loci containing important developmental genes show patterns of association between CNE density and topological organisation similar to those seen in humans (e.g., *hth*, *pros*, *CG34114* (Supplementary Fig. 6b) and the Antennapedia complex (Supplementary Fig. 6c)). Vertebrate homologues of these genes (e.g., *MEIS1/MEIS2* (*hth*) and *PROX1* (*pros*), *HOX* (*Antp*), *SOX1/2/3* (*soxN*)) are located in regions of extreme noncoding conservation (Fig. 1a and Supplementary Figs 2b, 6a) in hg19-galGal4 comparisons, which are highly predictive of the span of interactions observed in Hi-C (Fig. 3). This relationship is supported in a third Metazoan phylum: at the *Six* gene loci in the sea urchin *Strongylocentrotus purpuratus*, the regulatory landscapes defined by the span of promoter contacts identified using 4C[45] closely correlates with lineage-specific CNE density at this locus (Supplementary Fig. 7b).

These results show overwhelming evidence for a strong concordance between CNE density and topological organisation in the genomes of vertebrates and arthropods; with limited evidence available for echinoderms. These phyletic groups shared a common ancestor at ~560 Mya, and while CNEs in these phyla show no conservation between them, they are highly predictive of the extent of the regulatory domains of homologous genes in both lineages. This association is observable at individual loci and genome-wide, independently of how topological organisation is represented.

**TADs associated with GRBs exhibit distinct genomic features**. Genomic regions have previously been classified into TADs, inter-TADs and TAD boundaries based on their size and interaction structure[19]. Several genomic features potentially involved in delimiting TAD boundaries have been identified including gene density, CTCF binding and the distribution of repetitive elements. We have previously reported that GRBs do not cover the whole genome and that many genes fall outside of regions with high levels of noncoding conservation[13, 14],

suggesting differences in their structural and epigenetic organisation. Therefore, we investigated whether TADs containing high levels of extreme noncoding conservation (GRB-TADs) corresponded to a distinct functional subgroup of TADs compared to those lacking evidence of this type of conservation (non-GRB-TADs) (Supplementary Fig. 8a and Supplementary Table 3) (see Supplementary Methods).

Previously, it has been found that the regions surrounding key developmental genes are depleted of transposons[46], suggesting that the regulation of these genes is sensitive to insertions. Indeed, GRBs sharply define regions depleted of transposons, with strong increases in the density of SINEs in their flanking regions (Fig. 4 and Supplementary Fig. 8b). Changes in SINE density have previously been found to be associated with TAD boundaries[19]. Regions associated with GRBs are associated with lower levels of SINEs compared to all TADs and non-GRB-TADs (Fig. 4a), reflecting selective pressure on both the syntenic and regulatory conformation of these loci. There was no strong depletion of other types of retrotransposons in these regions (Supplementary Fig. 8b, c). Therefore, changes in SINE density at TAD boundaries is indicative of transitions between TADs under high negative selection against insertions and TADs which are not under this form of constraint.

In *Drosophila*, Filion et al.[47] used DamID to map multiple DNA-binding proteins and histone modifications in *Drosophila*, classifying chromatin into five putative states/colours (see Supplementary Methods). Using this classification, dm3-droMoj3 GRBs show the presence of transcriptionally silent regions (*black*), Polycomb-repressed chromatin (*blue*) or regulated euchromatin (*red*) within them (Fig. 4b and Supplementary Fig. 8d), along with a clear depletion of constitutive heterochromatin (*green*). Therefore, the majority of GRBs in Kc167 cells are either silent or largely repressed by Polycomb. The domains of *black* and *blue* chromatin closely correspond to the extent of GRBs, with the edges of GRBs showing evidence of enrichment of constitutively active (*yellow*) chromatin. This enrichment for *yellow* chromatin is analogous to the presence of ubiquitously expressed/house-keeping genes at TAD boundaries as reported previously[19]. The pattern of regulated chromatin at GRBs is concordant with the expected expression pattern of GRB target genes, which are repressed by Polycomb in the majority of tissues and marked with *red* chromatin when active. This suggests that GRB-TADs are coherent with respect to their chromatin state and correspond to demarcated regions of regulated chromatin, which are often flanked by domains of constitutively active chromatin.

Next, we examined the relationship between the boundaries of GRB-TADs and the patterns of CTCF binding across multiple mouse tissues[48]. CTCF binding is depleted inside GRBs identified between mouse and chicken (Fig. 4c); however, there is enrichment of CTCF peaks at the boundaries of GRBs, similar to the observed enrichment at TAD edges (Supplementary Fig. 9a)[19]. This enrichment is more prominent at the edges of TADs that overlap GRBs than at the edges of the GRBs themselves. This is expected from the GRB model and method of detection: since all CNEs of a GRB are expected to be inside the TAD that coincides with the GRB, the estimate based on CNE density is expected to be slightly shorter than the TAD span, depending on the distance between the outermost CNEs and the physical TAD boundary. It also explains the one-sided tail between CTCF signal and the estimated GRB boundary—the GRB-based boundary estimate is expected to be close to the TAD boundary, but always inside of it.

Importantly, CTCF peaks at GRB boundaries and TAD boundaries are enriched for constitutive CTCF binding sites ($p = 4e{-}11$, binomial test) and depleted for cell type-specific CTCF binding ($p = 1e{-}8$, binomial test) (Fig. 4d), in agreement

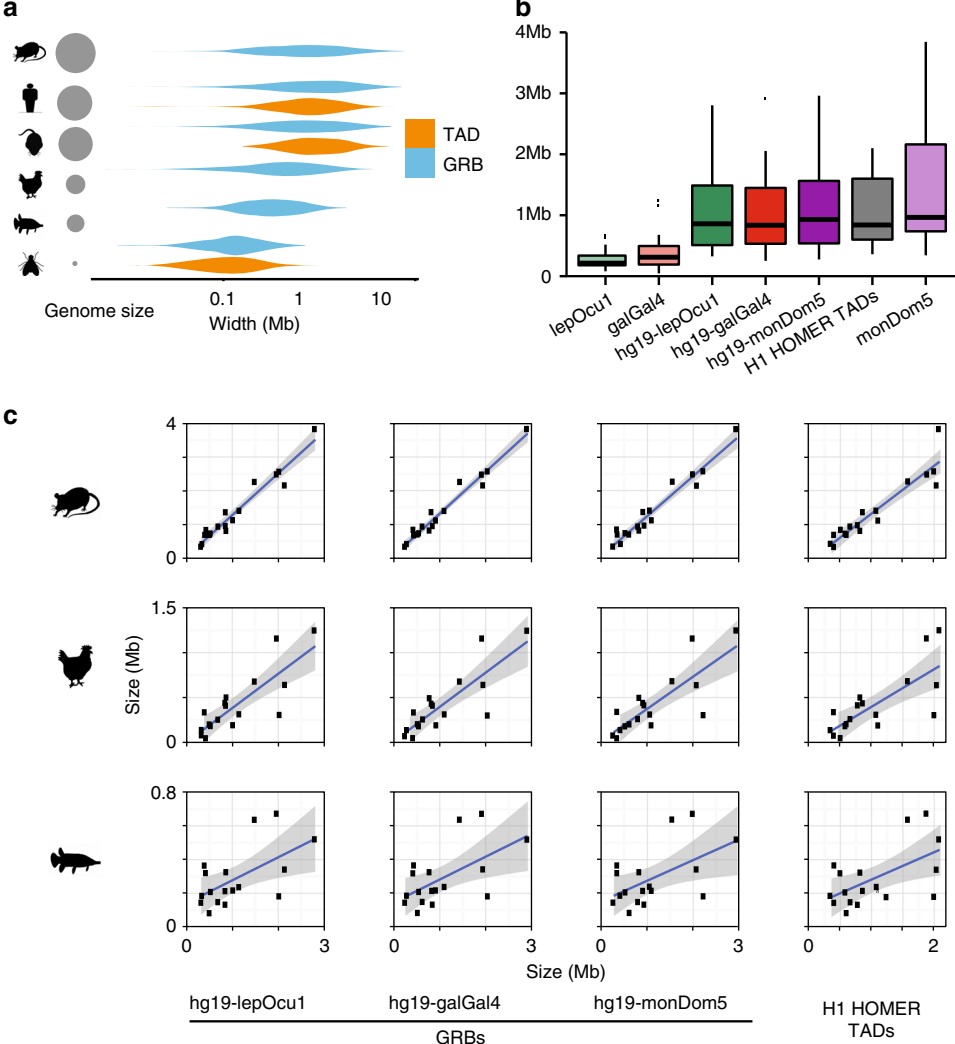

**Fig. 5** The span of CNEs in various species is predictive of genome size and TAD size. **a** Distribution of genome size, TAD size and GRB size in opossum, humans, mouse, chicken, spotted gar and *Drosophila*. **b** Sizes of 17 stringent GRBs identified in various species shows direct expansion and growth in line with that expected given differences in genome size. **c** Correlation of sizes of GRBs identified in one species compared to other species and to the corresponding TADs in humans finds that the size of a GRB identified in other species is highly predictive of the size and scope of regulatory domains in other species

with the invariance of TAD structure across cell types. CTCF peaks at GRB-TAD boundaries are enriched for constitutive CTCF binding ($p = 0.02$, binomial test), while non-GRB-TADs show no significant enrichment for constitutive binding ($p = 0.11$, binomial test), suggesting that the boundaries of GRB-TADs may be more consistent across different cell lines and tissues. This supports CTCF as being involved in the organisation and demarcation of these TADs, and the importance of insulating these domains containing developmental regulators from interactions with noncognate regulatory elements[22, 45, 49].

Next we showed that GRB-TADs are significantly longer ($p < 1e-6$, permutation test) than non-GRB-TADs in both humans (Fig. 4e) and *Drosophila* (Supplementary Fig. 9b), regardless of the method used to identify TADs. Despite their larger size, GRB-TADs contain fewer genes than non-GRB-TADs (Fig. 4g and Supplementary Fig. 9c). This is expected as GRBs are often associated with gene deserts in vertebrates[50] and larger introns and intergenic spaces in *Drosophila*[3]. The combination of these features suggests that these regulatory domains are larger to accommodate the numerous elements needed to regulate target genes during development. In addition, examination of their

directionality indexes showed that GRB-TADs are generally the strongest TADs in both the human ($p < 1e-6$, permutation test, Fig. 4f) and *Drosophila* genome ($p < 1e-6$, permutation test, Supplementary Fig. 9d). This suggests that GRB-TADs have higher levels of self-interaction and are more insulated from neighbouring regions compared to non-GRB-TADs.

Since GRB-TADs are gene sparse and their target genes are inactive in most tissues, we expect most GRBs in a given cell type to be associated with compartment B[31]. Indeed, regardless of cell lineage, GRB-TADs were found to be preferentially located within the B compartment (Fig. 4h and Supplementary Fig. 10a). Dixon et al.[23] reported that TADs that change compartment show concordant changes in expression of their constituent genes during development/differentiation. We found a clear enrichment of GRB-TADs in the set of TADs that switch compartment in one or more of the cell types investigated compared to non-GRB-TADs, which appeared to be more stable ($p = 1.35e-06$, Fisher's exact test, Fig. 4i and Supplementary Fig. 10b). At several GRBs, a change in compartment was associated with the change in expression state of the GRB target gene (e.g., *ZEB2*, *OTX2* and *SOX2*, Supplementary Fig. 10d–f). Intriguingly, while the target

gene showed concordant changes between expression and compartment, nearby bystander genes showed no evidence of this relationship. For example, while ZEB2 (Supplementary Fig. 10d) showed high expression in mesenchymal stem cells (MS), in line with its presence in compartment A, bystander genes within this GRB exhibited little or no change. This was also apparent at the OTX2 GRB (Supplementary Fig. 10e) in mesendoderm (ME) and neural progenitor cells (NP), and at the SOX2 GRB in NP (Supplementary Fig. 10f). This relationship between GRB-TADs and compartments further confirms that these regions represent the regulatory domains of developmental genes.

Several TADs not associated with GRBs identified using human–chicken comparisons nevertheless have features associated with GRB-TADs and contain developmental regulators. At a number of loci, human–chicken CNEs are present but were not effectively clustered by our approach; however, several loci lack CNEs identifiable between human and chicken. The CNTNAP4 locus (Supplementary Fig. 11a) is largely devoid of CNEs identified using chicken, but shows limited conservation between humans and spotted gar and high levels of noncoding conservation in comparisons involving dog (canFam3) and mouse. It appears that this regulatory domain is conserved over vertebrates, with the evolutionary patterns of its constitutive CNEs suggesting that these elements have been lost and gained in a lineage-specific fashion[7]. Investigating those non-GRB-TADs that switched compartment during differentiation identified several loci that exhibited high levels of noncoding conservation in comparisons with more closely related species (e.g., NECTIN3 and protocadherin alpha/beta clusters). Analysis of this set of genes revealed a strong enrichment for genes involved in cell–cell adhesion (Supplementary Fig. 10c). This suggests that these regions may represent loci which have undergone recent regulatory innovation[51], or whose regulatory elements are subject to high turnover[7].

These results support the idea that TADs associated with developmental genes and high levels of noncoding conservation have a distinct set of features indicating differences in their topological organisation and demarcation. We conclude that most TADs called in CNE-free and gene-dense regions of the genome exhibit reduced directionality of interactions compared to those associated with high levels of noncoding conservation, which may represent a distinct class of regulatory domain.

**The sizes of GRBs and TADs scale with genome size**. Next we investigated whether there was a relationship between genome size, TAD size and GRB size. Previously it has been shown that clusters of CNEs are more compact on average in Drosophila than in mammals[3], and that they scale with genome size in fish[52]. As expected, TADs and GRBs were larger in species with larger genomes (Fig. 5a). Therefore, we investigated whether the genomic locations of CNEs could provide information on the expansion and evolution of TADs in vertebrates.

We investigated a stringent set of 17 human GRBs which accurately predicted the boundaries (i.e., both edges within 120 kb) of the same set of TADs regardless of which species was used to identify GRBs. These include the GRBs containing MEIS1 (Figs 1, 3), IRX3 (Supplementary Figs. 2, 3) and other developmental genes such as PBX1, OTX1 and LMO4. The location of the homologous CNEs was used to predict an orthologous set of GRBs in spotted gar, chicken and opossum. In spotted gar this set of GRBs was 24% of the sizes of GRBs predicted in humans using the same set, with the ratio of genome sizes being 28%. This relationship was also observed in chicken (median GRB size ratio 37%, genome size ratio 33%) and opossum (median GRB size ratio 103%, genome size ratio 113%). Overall, the size of these regions in one species was highly

predictive of the size of the region in another (Fig. 5c). The ratios of these numbers and their strong linear relationship suggest that this set of domains has undergone expansion at a comparable rate to genome growth, and that the distribution of CNEs in one species could be used to predict the location of GRBs and their associated TADs in another.

However, this pattern of expansion and evolution was not the only one observed: some regions exhibit strong conservation of one GRB boundary whilst showing expansion at the other, e.g., HLX locus (Supplementary Fig. 11b), while others have a core highly conserved region in comparisons with spotted gar which is expanded in dog and opossum comparisons, e.g., CNTNAP4 (Supplementary Fig. 11a). These observations support the idea of multiple mechanisms involved in the evolution of regulatory domains in Metazoa including lineage-specific expansion of TADs, recruitment of neighbouring TADs and recruitment or turnover of regulatory elements both within these domains and at their edges.

**Discussion**

In this work, we show that the span of clusters of CNEs, known as GRBs, is predictive of the span of a subset of TADs in both humans and Drosophila. These sets of TADs, referred to as GRB-TADs, are some of the largest, strongest and most gene sparse in humans and Drosophila and show distinct patterns of retrotransposon density and CTCF binding. Regions containing homologous developmental genes are associated with the same type of conservation and structure in humans and Drosophila. Not only are regulatory elements in these regions under intense selective pressure, but the association between the boundaries of GRBs and TADs also suggests that the basic 3D structure of these loci has existed over hundreds of millions of years of evolution, at least back to the ancestor of chordates and arthropods. At even more distant timescales, the phenomenon of microsynteny has existed since early Metazoa and is apparent across bilateria[17, 53]. The syntenic relationship between RUNX2 and SUPT3H (Fig. 3b) is conserved between humans and sponges[54], and the GRB containing the Iroquois and Sowah genes is conserved across a wide range of bilaterians, apart from tetrapods[55]. This correspondence suggests that the regulatory domains around developmental gene paralogues created by whole-genome duplications (WGDs) are ancestral and not the result of convergent evolution[56], and may have existed since the origin of Metazoa.

The striking concordance between the distribution of CNEs and topological organisation has far-reaching implications for understanding the nature and evolution of long-range regulation at developmental loci. Combining the concepts of GRBs and TADs leads to a model of regulatory domains with stronger predictive and explanatory power than either concept alone (Supplementary Fig. 12). The GRB model provides a framework for long-range regulation, in which the majority of regulatory elements within a GRB are dedicated to the control of its target gene[9, 11], with other genes not responding to long-range regulation despite being located close to regulatory elements[8, 57]. Target and bystander genes appear to exhibit distinct features that may help to explain this specificity (see Akalin et al. for a list of putative target genes)[14, 15]. Topological organisation data provide more precise boundary estimates for these regions, including the ability to separate adjacent GRBs, and contributes information about the stability of the organisation of these domains and regulatory interactions within them across cells and tissue types. Therefore, a TAD enriched for extreme noncoding conservation is not representative of the regulatory domain of all of its constitutive genes, but primarily corresponds to the regulatory domain of the target gene under long-range regulation.

The lower density of active genes and increased likelihood of Polycomb repression at GRBs may explain some of their topological features. Since GRBs represent the regulatory domains of developmental genes, in any one tissue or developmental stage most GRBs will be inactive and marked by Polycomb/H3K27me3[58]. The degree of intermixing of chromatin between genomic domains depends strongly on their epigenetic state, with Polycomb-repressed chromatin showing little or no spatial overlap with active or inactive chromatin[59]. It has been suggested that regions of active chromatin within TADs interfere with the packaging of chromatin, disrupting TAD formation and leading to less compact TADs, or fragmentation into smaller TADs[60]. The enrichment of active euchromatin at the boundaries of these regions may reflect the formation of barriers.

The conservation and divergence of CTCF-binding sites is thought to play important roles in the evolution of regulatory domains[25, 45]. Constitutive CTCF sites are enriched at GRB and GRB-TAD boundaries, suggesting these regions are consistently insulated from neighbouring domains. The enrichment of CTCF at GRB and GRB-TAD boundaries, combined with the strength of interactions associated with GRBs, further suggests that these regions do not strongly interact with elements in adjacent domains. This is in addition to the association of GRBs with Polycomb-repressed chromatin and depletion of active euchromatin, which also promotes insulation from neighbouring regions, as described above. These features are reflective of the multiple mechanisms involved in insulating target genes from ectopic regulation by elements in neighbouring domains. It is highly probable that this type of restricted topological structure arose early in Metazoan evolution as a result of strong selective pressure to prevent this.

GRBs and their constituent CNEs are defined by sequence conservation alone, which is stable across all cell types, with TADs having been found to be largely invariant across cell lines and between species[19]. Even though our methods for estimating GRBs have limitations with respect to their precision and coverage, we have shown that the distribution of CNEs can nevertheless serve as an excellent proxy for the extent of a functionally distinct subset of TADs in both humans and *Drosophila*. Therefore, clusters of CNEs identified between evolutionarily distant species could be used to infer regulatory domains and predict their topological organisation in species lacking Hi-C data. TADs are also organised into inactive and active chromatin compartments, and the assignment of TADs to compartments can vary across cell types[19, 20, 23, 31]. GRB-TADs are preferentially located within compartment B across all of the lineages investigated, although they are more likely to switch compartments than those TADs lacking extreme noncoding conservation. This is consistent with previous work showing that GRBs represent regions that encompass regulatory elements of specific developmental regulators, which are repressed during the majority of developmental stages and cell types and only expressed in a limited subset.

A remarkable property of TADs and GRBs is that they expand and shrink along with the entire genome, suggesting that the 3D organisation of regulatory loci is robust towards gain and loss of DNA between its constituent CNEs, even though insertion of repetitive elements is disfavoured. Previously, we observed that GRBs are on average much more compact in *Drosophila* than in mammals[3], and that they also scale with genome size in fish genomes[52], and as previously observed TADs are smaller in *Drosophila*[28], concordant with our hypothesis. Duplicated loci containing developmental regulators are more likely to be retained after WGD[61]. Therefore, since tetrapod lineages have undergone two WGDs, a larger number of developmental regulators and associated GRBs are expected compared to arthropods, which is confirmed by our analysis.

The expansion and shrinkage of TADs and the turnover of regulatory elements within them is likely to be an important mechanism in metazoan evolution[62, 63] and responsible for differences in organismal complexity[7, 64–67]. The disruption of TAD boundaries perturbs spatial organisation, enhancer–promoter interactions and the expression of target genes[22, 45]. Deletions within TADs can lead to changes in enhancer–promoter interactions, in some cases causing disease[68, 69], which suggests some level of selective pressure against such re-arrangements. The depletion of retrotransposons within GRB-TADs suggests that their insertion within this type of regulatory domain is under intense negative selection. This pressure may be due to the ability for retrotransposons to create new *cis*-regulatory elements[70], potentially perturbing the organisation of interactions within a regulatory domain, leading to ectopic expression and resulting in a negative effect on fitness in the majority of cases[71]. In future, analysis of the evolutionary dynamics of CNEs and genomic rearrangements within TADs in multiple species will help to provide insights into their evolutionary dynamics.

Because of the unknown reason for the extreme conservation of CNEs, their function as enhancers, and that their distribution closely follows the span of the TADs around genes known to be involved in long-range developmental regulation, it is tempting to speculate that CNEs are somehow directly involved in the chromatin folding of TADs, precisely arranging promoters and enhancers in 3D space. Indeed, just like TADs, the spatial proximity of promoters and developmental enhancers seems to be stable across different cell types, regardless of the activity of either[72]. Currently, there is no evidence from sequence analysis that CNEs are involved in sequence-mediated interactions, and their role in chromatin folding remains an open question.

We conclude that this subset of TADs, which are associated with high levels of noncoding conservation, are functionally distinct, evolutionarily ancient 3D structures which represent the regulatory domains of key genes involved in embryonic development and morphogenesis. However, the spatial correspondence of GRBs and TADs does not offer immediate suggestions for the origin of extreme noncoding conservation. Just like other potential sources of selective pressure acting on these elements, current models of genome folding do not include a mechanism that could account for this level of selective pressure on elements within TADs. The main findings of this paper may help with formulating new hypotheses by focusing on their potential roles within TADs.

## Methods

**Identification of conserved CNEs.** Conserved CNEs were generated by examining pairwise BLASTZ net whole-genome alignments[73] for regions with a high percentage identity over a defined number of base-pairs. For each comparison, both of the relevant nets (from the perspective of each species) were scanned. Elements overlapping exonic and repetitive repeats were removed. This set of elements was then aligned against the genome using BLAT to remove elements that mapped to more than four locations in vertebrates. The resulting set of CNEs was then smoothed using a sliding window (300 kb for vertebrates and 50 kb for *Drosophila*) to generate CNE densities, as was originally used for ANCORA browser[33]. As the evolutionary distance between two species increases it becomes more difficult to identify CNEs, and therefore it is required that less stringent thresholds are used.

**Identification of GRBs.** GRBs were generated by identifying regions of the genome that contain a high density of syntenic conserved CNEs. CNE densities were generated using CNEs that showed a high level of percentage identity (i.e., at least 70–96%) over a number of base pairs (i.e., 30 or 50 bp) between two species. These densities were partitioned into regions with high or low CNE density using an unsupervised two-state hidden Markov model (HMM)[74] (Supplementary Fig. 1b). This segmentation was performed 10 times, with the set having the best Akaike information criterion defined as the best model. All CNEs that were not present in an enriched region were removed. The remaining CNEs were merged using the distance between adjacent CNEs as a criterion. The sizes of gaps between individual CNEs for each chromosome were determined and

used to recursively split the genome into regions where the distance between adjacent CNEs was greater than a specified quantile of the gap distribution. Our experiments with this threshold suggested that as the evolutionary distance between the two species of interest increased, the quantile used needed to be decreased. We used 0.98 for hg19-monDom5 comparisons, 0.98 for hg19-galGal4, 0.93 for hg19-lepOcu1 and 0.97 for dm3-droMoj3. These parameters were determined empirically by investigating the ability of our predictions to recapitulate known boundaries of GRBs. Following this, putative regions were split by the chromosome that they originated from in the query species to generate discrete regions of conserved synteny. Regions that did not contain a protein-coding gene were merged with adjacent regions if they were within 300 kb (for humans/mouse) or 50 kb (for *Drosophila*). All remaining regions not containing a protein-coding gene or having <10 CNEs was discarded, resulting in a set of putative GRBs.

Our initial investigations found that segmenting CNE density using either a HMM or the distance-based criterion alone had difficulties generating a robust set of putative GRBs. The HMM approach had a tendency to merge adjacent GRBs, while the distance-based clustering appeared to be very sensitive to isolated CNEs and tended to overestimate the span of GRBs. By combining both methods together, the impact of the problems associated with each method was reduced.

Two main problems can occur with the prediction of GRB span from CNE density. If the synteny between two adjacent GRBs is conserved in the species used for comparison and the GRBs are extremely close together, there is no information from the CNE density alone that would enable their separation during clustering. For that reason, we expect that a fraction of estimated GRB spans will contain multiple adjacent GRBs, and predict that this will be more prevalent with the longest span predictions. Alternatively, since the density of CNEs along a GRB is non-homogeneous, a single GRB can be split into two or more putative regions. We expect this will be more prevalent in predictions of smaller GRBs.

**Processing of Hi-C data sets**. The set of TADs generated from an experiment is dependent on both the experimental protocol (i.e., restriction enzyme, sequencing depth) and the processing techniques used (i.e., bin size, algorithm). Hi-C interaction data sets for human were obtained from Gene Expression Omnibus (GEO; GSE52457), for H1-ESC (H1), MS, ME, NP and trophoblast-like[23]. These reads were iteratively aligned[75] using bowtie[76] against hg19. Reads mapping to chrM and chrY were removed from the analysis. The resulting aligned reads were binned using a variety of bin and window sizes, with a bin size of 20 kb and a window size of 40 kb appearing to generate a robust set of TADs. TADs were identified using both HOMER[34] and the TAD calling pipeline (HMM_calls) proposed by Dixon et al.[19]. Mouse ESC Hi-C data were downloaded from GEO (GSE35156) and processed using the same pipeline as for humans. Hi-C for *Drosophila* whole embryo Hi-C data[28] were obtained from GEO (GSM849422) and processed using the same pipeline. Directionality matrices for *Drosophila* Hi-C were generated using HOMER with a bin size of 10 kb and a window size of 20 kb. Due to the low number of reads mapping uniquely to heterochromatic chromosomes (i.e., chr3RHet, chr2LHet), these chromosomes were discarded from further analyses. TADs predicted to span across centromeric regions were removed.

The strength of TADs was defined as the sum of the absolute directionality indexes within a TAD normalised to the length of the QTAD in kilobases.

High-resolution GM12878 Hi-C data were obtained from GSE63525[20]. Contact domains which overlapped by at least 60% were collapsed to generate a set of *outer-most* domains.

Compartments were identified by performing principal component analysis on the Hi-C interaction matrix and investigating the first principal component. TADs were classified as A or B given at least 60% of locations within them were either positive or negative, respectively. A single gene was classified by examining at 5 kb window around its promoter and classifying it as belonging to the A or B compartment using the same criteria.

**Visualisation of the relationship between GRBs and TADs**. To visualise the relationship between the identified GRBs and TADs, we produced heatmaps of genomic regions centred on the GRB and ordered by GRB size, in which the GRBs and any features that correlate with them show a characteristic funnel shape. To show the TAD data for the GRB regions shown on the heatmap, we used Hi-C directionality index (positive/red when this region is preferentially interacting with regions downstream, and negative/blue when this region is preferentially interacting with regions upstream; one TAD is typically a red region followed by similar-sized blue region).

**GRB-TADs vs. non-GRB-TADs**. TADs were classified as GRB-TADs and non-GRB-TADs based on their overlap with GRBs. In humans, TADs with >80% overlap with a single GRB were assigned as GRB-TADs, TADs with <20% overlap with a GRB were assigned as non-GRB-TADs and any TAD that overlapped more than one GRB or had an overlap percentage between 20 and 80% was screened out of this analysis. For *Drosophila*, a TAD with >60% overlap with a

single GRB was designated as a GRB-TAD, with TADs showing <25% overlap with a GRB defined as a non-GRB-TAD. These thresholds were determined by investigating the distribution of percentage overlap between GRBs and TADs genome-wide.

The significance of the differences between median TAD width, gene density and strength were calculated empirically by randomly permuting the labels of TADs 1 million permutations to generate a null distribution for each statistic.

**Repetitive elements**. Repetitive elements were obtained from the UCSC (University of California, Santa Cruz) Genome Browser[77] for hg19 and dm3. We investigated both the distribution of different classes of transposons (LINEs, SINEs, DNA or LTRs). For visualisation using heatmaps the average coverage of elements within 5 kb bins in a 8 Mb window around the centre of hg19-galGal4 GRBs was calculated. For average profiles, retrotransposons per 10 kb were calculated by counting the number of retrotransposons occurring in overlapping 10 kb windows, with a step size of 1 kb, across the human genome.

**CTCF binding at GRBs and TADs**. For this analysis, we used GRBs called using mm9-galGal4 CNEs at a threshold of 70% over 50 bp. CTCF chromatin immunoprecipitation sequencing data were obtained from mouse ENCODE[48] for 17 cell lines and tissues. Reads were aligned to the mm9 genome using bowtie[76] and peaks called using MACS2[78], with the first input replicate for each sample used as the control. Where replicates were available, the intersection of peaks called on different replicates was used for the final peak set.

A consensus set of CTCF peaks was calculated by resizing all peaks to a width of 400 bp and taking the union of peaks across all 17 samples (average CTCF peak size across all data sets investigated was 404 bp). Peaks were scored for the number of samples they occur in. CTCF peaks per 10 kb tracks were calculated using the consensus peak set and counting the number of peaks occurring in overlapping 10 kb windows, with a step size of 1 kb, across the mouse genome.

CTCF peaks within 10 kb of GRB and TAD boundaries were assigned to the boundary, and classified as 'specific' if they were present in 1–2 samples, 'constitutive' if they were present in 16–17 samples and 'intermediate' otherwise. Enrichment was calculated relative to the proportion of these categories in the consensus peak set, and *p*-values calculated for each category using a two-sided binomial test. Additionally, we investigated the effect of using different distance thresholds for assigning CTCF sites to GRB boundaries. Regardless of the choice of threshold used (10–120 kb in steps of 10 kb) the reported enrichments remained stable.

**D. melanogaster chromatin states**. *Drosophila* chromatin state data were downloaded from GEO (GSE22069). Filion et al.[47] identified five chromatin states in *D. melanogaster* Kc167 cells. Each of these states was found to have specific characteristics, including transcriptional activity, replication timing and biochemical properties. *Black* chromatin was associated with transcriptionally silent developmentally regulated chromatin, *green* with classical heterochromatin and *blue* with chromatin bound by PcG proteins.

*Red* chromatin was associated with early replicating euchromatin, a high density of regulatory elements and the presence of genes under complex, long-distance regulation, whereas *yellow* chromatin was associated with late-replicating euchromatin and genes with ubiquitous expression patterns (house-keeping genes). Heatmaps were generated by dividing a 1 Mb region around the centre of dm3-droMoj3 GRBs into 1 kb bins, with each bin assigned the most common chromatin colour within it.

**Comparisons of genome, TAD and GRB size**. A set of 17 human GRBs that accurately predicted the boundaries (i.e., both edges within 120 kb) of the same set of TADs regardless of the species involved (i.e., using monDom5, galGal4, lepOcu1) were identified. For each GRB, the location of the constitutive CNEs in the reference species was identified and used to generate a putative GRB in that species. Any CNE that was found to be present within the GRB but originated from a separate part of a chromosome was removed.

**Code availability**. The code that reproduces analyses from the manuscript is available at https://github.com/ComputationalRegulatoryGenomicsICL/tad_cnes_harmston2017

**Data availability**. Putative GRBs generated using comparisons of hg19-galGal4 (70%/50 bp) and dm3-droMoj3 (96%/50 bp) are available as Supplementary Data 1, 2, respectively. A trackhub containing all of the data presented in this analysis is available at http://trackhub.genereg.net/harmston2016/harmston2016.hub.txt

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

## Acknowledgements

We thank Piotr Balwierz, Anja Barešić, Sarah Langley, Dimitris Polychronopoulos and Owen Rackham for their comments on the manuscript. N.H., E.I.-S. and B.L. are supported by the Medical Research Council UK. G.T. is supported by the Wellcome Trust project P55504_WCMA. G.T. and B.L. were supported by EU project ZF-Health (FP7/2010-2015 grant agreement no. 242048). M.P. was supported by the Dean's PhD Scholarship, Faculty of Medicine, Imperial College London.

## Author contributions

N.H., E.I.-S., G.T. and M.P. performed computational analyses. N.H. and B.L. conceived and designed the study. N.H., E.I.-S. and B.L. wrote the manuscript.

## Additional information

**Competing interests:** The authors declare no competing financial interests.

