## [Peer Review File · Nature Communications]

Reviewers' comments:

Reviewer #1 (Remarks to the Author):

Harmston and coworkers have taken advantage of the deeply sequence-conserved developmental CNEs of metazoans to mark syntenic chromosomal regions so they can be compared among genomes diverged for over 400 million years. That is: the experimental design of this work is excellent. Because CNEs are automatically functional, this entire project takes on meaning. This reviewer would like to thank the authors for including an example genomic regions (Figure 3 and SIs) so the meaning of GRBs could be visualized in relationship to real genes. That a good portion of the metazoan genome is organized into TAD-GRBs with conserved, insulated boundaries, and that these regions occur in the chromatin compartment that is largely "off" makes a lot of sense. It is interesting and meaningful that tetrapods and arthropods have inherited gene-region-specific TAD-GRB organization from an ancient ancestor.

This reviewer also found it reassuring that so many TAD boundaries coincided with genetic and epigenetic features. Hi-C is clearly measuring meaningful DNA juxtapositions. If there are mutants that might be expected to disrupt the sort of looping measured here by Hi-C (like MORC-negative mutants in plants), they would have made a great control. This is a comment, not really a suggestion.

This reviewer did not understand the last sentence; clarity might help this paper reach a broader audience. "Current models of genome folding do not include a mechanism that could account for this level of selective pressure on elements within TADS." I was not aware that the extreme and phylogenetically deep sequence conservation of CNEs was the issue here. Naively, perhaps, I assumed that these CNE sequences served multiple (at same sequence) binding roles constraining diversity (like binding a TF and binding a looping protein, for example). Genome folding, on the other hand, like part of packaging, does seem to be a considerable mystery, especially in light of the C-value "paradox", here restated as the TAD-expansion paradox. This reviewer would appreciate some discussion of the hypothetical functions that might be provided by the balanced distribution of CNEs within TADs of different length, as if the evenness of the distribution was under selection without regard to total length; folding seems to fit this sort of phenomenon.

The METHODS seem to be state of the art.

Minor. Figure 4 legend, l4. Perhaps replace "...against repeat element insertion." with "... against the retention of repeat element insertion."

Reviewer #2 (Remarks to the Author):

Harmston et al. have conducted a genome-wide research where they try and assess the correspondence between TADs and evolutionarily conserved regulatory landscapes. To this aim, they first identify conserved non-coding elements (CNEs) present in various vertebrate genomes and displaying different degrees of synteny with the human genome. Subsequently, they group these CNEs into genomic regulatory blocks (GRBs) following different methods. They then analyse the distribution of these GRBs in the context of topologically associating domains (active or inactive) as well as with other elements like the presence of CTCF. The conclusions of this study generally support those of previous works, i.e. that TADs often include regulatory landscapes with a number of enhancers and their target genes, and that TADs are usually kept throughout evolution. These results are in many cases somehow redundant and it was not always easy to us to see the added value of this particular piece of work. Although the initial hypothesis is well accomplished, there is no challenge of previous models or, alternatively, a proper experimental approach to either confirm or reject the presented results. Therefore, we support publication of

this paper with a moderate enthusiasm only and would rather see it appearing in a more 'technically oriented' forum.

While the question to know whether or not the presence of CNEs is in a way reflected by the existence and/or structure of TADs is valid, it is now well established that dense regulatory landscapes –which contains many CNEs- do generally coincide with TADs structures (the question of causality is however still unclear). Therefore, we are not fully convinced by the novelty of this study, when taken at a general level. In addition, there are some technical concerns about how the results were generated and how their graphic and statistical representations are formalized, as well as some conceptual problems, to our opinion. Some of these issues are listed below.

- The funnel shaped density maps are used extensively. Should this article be targeted to a wide readership, some more information should be added either on the axis or in the figure legend in order to make understanding easier. Also, the work lacks proper statistical analysis: we do not think it is appropriate to talk about concordance simply because two plots have similar shapes.
 - The authors like to emphasize the good correspondence between TADs and GRBs. TADs are chromatin regions originally defined by their structural interaction profiles. Previous studies have already correlated regulatory landscapes (including enhancer-promoter interactions) with the distribution of TADs. In their approach, the authors make the assumption that conserved regions display presumed regulatory potentials. This referee has some difficulty to see where the novelty stands in this proposal, with respect to previously published papers (including reviews).
 - If the aim of this work is to compile conserved regions with regulatory potential, several epigenetic marks (e.g. H3K4me1) could have been used instead of identifying such regions, as well as to remove promoters that would complicate the interpretation. In fact, the authors address this question of whether or not epigenetic marks could be a valuable parameter to identify GRBs. However, the use of H3K27ac datasets from different cell lines is likely not suitable for this purpose, since many genes and/or their enhancers may not be active in the cell lines analysed. To overcome this problem, the authors may advantageously use other parameters, which could be taken as hallmarks of poised enhancers like p300 or K4me1 or ATAC-seq datasets. In this case, enrichment should be detected even in the absence of transcription.
 - The authors rightly argue that regions positive for H3K27ac do not allow on their own to define GRB with accuracy, because they are also enriched in promoter regions. This limitation could be fixed by removing in silico either all TSS surrounding regions, or all loci covered by H3K4me3.
 - While a big proportion of GRB overlaps with a single TAD, an important proportion also encompasses more than one, sometimes even more than two TADs. It is not clear to this referee as to why a definition of GRB based on synteny and density of H3K27ac would be a better predictor of GRBs than the TADs themselves. Also the majority (if not all) of the examples provided in this study correspond to developmental genes flanked by relatively large gene deserts/regulatory landscapes. It is thus not entirely clear whether this approach would also identify GRB in more gene-dense regions or, alternatively, in TADs enriched in housekeeping/non-developmental genes?
 - An important fraction of GRBs correlate with TADs, the authors say. Interestingly, they show that 'GRB-TADs' tend to have a lower density of transposons and that they switch from activity compartment A to B and vice-versa more than non 'GRB-TADs'. Several studies have correlated TAD borders with a number of features including the density of CTCF binding sites. GRB-TADs also seem to be more enriched in constitutively bound CTCF at their borders than non GRB-TADs. These observations are interesting indeed, yet should the identification of GRB be biased in favour of identifying GRBs in regions that are well organized into TADs (see point 1), this would naturally impact upon the interpretation of the results.
 - One conclusion of the paper is that GRBs are as good predictors as TADs of the existence of regulatory landscapes around given genes. Could the authors provide evidence that GRBs predict the presence of a regulatory domain as efficiently as Hi-C-defined TADs? Can GRBs be found also in inter-TAD regions? Examples would be nice to illustrate these two points.
 - The authors fall short in estimating which structures predated the other. Did GRBs appear before TADs were established and organized? Or the contrary? Is the former the result of the latter? Was

there a selective advantage to establish 'private' chromatin structures in order to prevent illegitimate enhancers-promoter contacts? Here the general lack of an experimental approach makes this sort of conclusions difficult, which seriously limits the impact of the study. Hi-C datasets (or 4C-seq datasets) on some of the compared species would definitely bring more power to the hypothesis.

Methodology

*For the identification of conserved non-coding elements, three different kinds of regions were excluded: exonic sequences, repetitive sequences as well as sequences present at least four times in the genome. Could this latter criterion exclude certain discrete binding sites for transcription factors?

*Also, in the processing of Hi-C datasets, why were heterochromatic chromosomes discarded from further analysis? The authors should justify their choice in the text.

*In their calculation of the expected number of GRB boundaries lying within a specific distance of a TAD boundary, the authors consider that if both GRB borders are within three Hi-C bins of a TAD border, they are taken as 'coincidental'. If the Hi-C data were binned at 20kb, this means that the 'coincidental region' is around 60kb (or 120kb if the window size is what the authors seem to take as a reference -see on line 602). However, looking at Fig. 2D, it seems that 50% of the GRBs analysed display borders at a distance of 250kb from a TAD border. Taking into account that TADs are 800kb large on average in mammals, 250 kb would represent more than a quarter of their length, while such difference is not appreciable in the heatmap graphs (Fig. 2A). Also, this would mean that a large proportion (around 50%) of GRB borders would not coincide with the TAD border according with the criteria used by the authors. Could the authors provide an estimation of the average distance between GRB border and TAD borders? Alternatively the authors could provide a summary table with the numbers of GRBs and TADs with and without overlapping borders and discuss this discordance?

*The classification GRB-TADs versus non-GRB-TADs is not easy to understand. A little scheme with a graphical depiction of the classification could be advantageously added as supplementary material. Also, there seems to be a mistake on line 545 in the description of non-GRB-TADs: 'TADs with less than a 20% overlap with a TAD were assigned as non-GRB-TADs...'. And how much overlap between two TADs was considered in order to discard them from the analysis? And consequently, how many TADs were classified as belonging to either one or the other category?

*Regarding repetitive elements, why were only SINEs used in the analysis?

*Regarding CTCF binding at GRBs and TADs, why was a width of 400bp established for the consensus peaks?

*The distance selected to define the boundary region (10kb) is also unclear. It seems that GRBs are often shorter than TADs (see comment above). Thus, the distance between GRB borders and TAD borders should be taken into consideration in order not to under-evaluate the relationship of CTCF with GRB borders. This may indeed have an impact upon the interpretation: if a given TAD evolved by containing important enhancers for developmental genes, some presumably constitutive architectural contacts may have evolved to limit the extension of enhancer activity to prevent aberrant expression of surrounding genes. In this context, TAD border may coincide closely to GRB borders and this section of the paper is not fully clear.

Minor comments

Line 107-114. Human versus Gallus. 819 GRBs when using 70% identity of 50bp CNEs. Human vs Gar. 719 when using 70% identity of 30bps CNEs. Why were different lengths of CNEs considered for each species? Is it because identifying CNEs becomes more difficult at longer evolutionary distance? This should be explained in the text.

Fig1A. The figure is divided into different panels showing: chromosome position in the top, CNEs density distribution and a smooth version of the same. Also, several bars representing the definite borders of these GRBs are shown. Although, it seems relevant to show the different degrees of concordance according to the level of assumed identity, the plotted information is redundant. The same results are plotted in almost two identical ways (discrete densities and smoothed densities).

Fig1B. The authors plot the distribution of all analysed human versus Gallus GRBs according to their lengths. It would be useful to better understand what is the average size of the GRBs. If the authors want to state that approximately 80% of the conserved GRBs are shorter than 3Mb in length, they should include this in the text.

Fig1D. This graph is generally difficult to understand. What is exactly plotted there? Were the different GRBs in each species first centred, ordered and then overlapped? Which set of hg19-galGal4 (70%-80%-90%) was used? It seems that some explanations come later in the text (line137-140), but this should be stated when introducing the figure.

In line 121, it is stated that the subset of conserved GRB boundaries include developmental regulators. Only MEIS1 and IRX3 are used as examples. A comprehensive list of these regulators would give more power to this affirmation.

Figure2. It would be interesting to see an overlay between the different dataset (GRB and Hi-C directionality index). By superimposing the two distribution heatmaps, the concordances and differences at the edge of GRBs would be better spotted.

Fig2A. Are the Hi-C data-sets good enough to properly map them at 5kb resolution? And why is the 5kb bin distribution added to the figure? Is there any extra information there?

Fig2C. It is well-established that TADs do not dramatically change between cell lines. Why then was the GRBs-TAD correlation supposed to change?

Several examples of GRBs that fit TAD distribution are given. However, two of these examples seem to point to the opposite direction. In line 198-201 and Fig3, the region encompassing TOX3 and SALL1 belongs to the same GRB, whereas the interaction matrixes clearly show a split distribution of contacts. In Fig S5, something similar occurs at the HoxD locus. How do these examples integrate the general conclusions?

Fig S6. There may be a risk to try to extrapolate genome wide a situation observed only at one locus.

The sentence: 'changes in SINE density at TAD boundaries is indicative of transitions between TADs under high negative selection against insertions and TADs which are not under this form of selective pressure' is unclear to us. Why is it not a local effect that can affect only boundary regions?

Line 671. Typo: Loess-smoothed.

Reviewer #3 (Remarks to the Author):

Harmston et al. present evidence that highly conserved noncoding sequences, or CNEs, are often located at the boundaries of topological association domains (TADs) containing developmental control genes. This is an important observation and certainly worthy of publication in Nat Comm. I have just a few minor comments for the authors to consider prior to publication:

1. It would be helpful for the authors to explicitly state what % of the ~3000 TADs in the human genome contain CNEs/GRBs at their boundaries. How do these number change with a sliding definition of CNEs and GRBs?
2. Do the particular human TADs containing GRBs coincide with regions of "deep synteny" described by Putnam et al. Nature 2008?
3. It might be useful to include a more in-depth analysis of the Hox clusters (e.g., Hoxd) since these include a number of notorious TADs that have been defined in functional terms.

Response to reviewers

We thank the reviewers for the constructive and generally positive comments on the manuscript. In response, we performed several additional analyses, rewrote parts of the text that the reviewers found insufficiently clear in the original submission, and added several points to the discussion. The largest additions and changes are as follows:

- We have added a discussion of the potential role of GRBs in genome folding (Reviewer #1).
- The supplementary data now contains funnel heatmaps for H3K27ac and H3K4me1 marks, both with and without signal from promoter regions. Unlike GRBs, these marks do not distinguish between TADs and their surroundings, just as our model predicts. (Reviewer #2)
- A new supplementary table (Table S1) is added, containing statistics on numbers and sizes of CNEs, GRBs and TADs, along with numbers corresponding to GRB-TAD/nonGRB-TAD classifications. (Reviewer #2)
- In addition to SINE elements, we have now included the heatmaps and average profiles showing the distribution of other classes of repeats in GRBs. The conclusion is that these other classes of repeats are not capable of delineating adjacent TADs or between TADs with high non-coding conservation compared to those lacking non-coding conservation (Reviewer #2).
- We have analysed existing 4C data and provided an in-depth view of the HoxD locus with respect to CNE and TAD content (Figure S7) (Reviewer #3).

We have responded to all comments in detail below.

We believe that the results of additional analyses further strengthened our original conclusions, demonstrating previously unreported high concordance between TADs containing developmentally regulated genes and genomic clusters of extreme non-coding conservation. This concordance is not revealed by poised, active or aggregated enhancer marks and represents an independent way to approximate the span of TADs that contain developmentally regulated genes. As such, it will serve as a starting point for the investigation of mechanism of chromatin folding into TADs and potential selection pressure patterns originating from the need to conserve it.

Point-by-point response

Reviewer #1:

Harmston and coworkers have taken advantage of the deeply sequence-conserved developmental CNEs of metazoans to mark syntenic chromosomal regions so they can be compared among genomes diverged for over 400 million years. That is: the experimental design of this work is excellent. Because CNEs are automatically functional, this entire project takes on meaning. This reviewer would like to thank the authors for including an example genomic regions (Figure 3 and SIs) so the meaning of GRBs could be visualized in relationship to real genes. That a good portion of the metazoan genome is organized into TAD-GRBs with conserved, insulated boundaries, and that these regions occur in the chromatin compartment that is largely “off” makes a lot of sense. It is interesting and meaningful that tetrapods and arthropods have inherited gene-region-specific TAD-GRB organization from an ancient ancestor.

This reviewer also found it reassuring that so many TAD boundaries coincided with genetic and epigenetic features. Hi-C is clearly measuring meaningful DNA juxtapositions. If there are mutants that might be expected to disrupt the sort of looping measured here by Hi-C (like MORC-negative mutants in plants), they would have made a great control. This is a comment, not really a suggestion.

We thank for the reviewer for this comment, and agree that the analysis of the effects of mutations of genome organisation factors on organisation at GRB-TADs would be a good next step to take based on hypotheses generated by the observations in our manuscript. Indeed, in the paper we discuss published experimental work which has confirmed that disruption of topological organisation at a locus with high levels of non-coding conservation and containing developmental regulators leads to developmental defects. (Lupiáñez et al. 2015). Lupianez et al. investigated the impact of perturbing CTCF binding at the boundaries of TADs containing the developmental regulators *WNT6*, *IHH*, *EPHA4* and *PAX3*, and found that this leads to developmental defects and ectopic gene expression. The region/TADs surrounding these genes has high levels of extreme non-coding conservation. We believe that this is independent evidence supporting the functional role of these domains suggested by GRB-TAD association and the importance of preventing ectopic enhancer usage by developmental genes.

This reviewer did not understand the last sentence; clarity might help this paper reach a broader audience. “Current models of genome folding do not include a mechanism that could account for this level of selective pressure on elements within TADS.” I was not aware that the extreme and phylogenetically deep sequence conservation of CNEs was the issue here. Naively, perhaps, I assumed that these CNE sequences served multiple (at same sequence) binding roles constraining diversity (like binding a TF and binding a looping protein, for example).

We have now tried to make clearer that the CNE conservation is both extreme and as of yet unexplained. Our review article (Harmston et al. 2013) is dedicated to this topic, and we expanded the last sentence to refer the readers to it, and to stress that we are still searching for the mechanism behind extreme non-coding conservation.

Here is the summary of the issue, treated in detail in the review: CNEs are DNA elements ranging in size from 30bp up to a couple of hundred bp, which is longer than individual transcription factor binding sites, and they show unusually high percentage identity across genomes of distantly related species. Current models of enhancer function and evolution cannot account for this level of conservation. For example, several CNEs exist that are longer than 200bp and 100% identical between human and rodents (Bejerano et al. 2004) The enhanceosome model of enhancer function requires only conservation of sequences for binding sites and the distances between them, but does not require, or explain, the observed pervasive strong conservation of sequences between transcription factor binding sites; indeed, it does not require perfect sequence conservation of the binding sites, either, since their recognition motifs are generally degenerate (Wasserman and Sandelin 2004) In order for overlapping motifs to be able to explain this level of conservation, there would have to be an extremely dense overlap of multiple functional TFBSs or nucleosome positioning signals, which has however not been observed in any of the available ChIP-seq or chromatin accessibility data. In short, we still lack adequate hypotheses for the origin of form of this selective pressure. The closing sentences of the paper now read “Just like other potential sources of selective pressure acting on these elements we discussed previously, current models of genome folding do not include a mechanism that could account for this level of selective pressure on elements within TADs. The main findings of this paper will help with formulating new hypotheses by focusing on their potential roles within TADs. ”

Genome folding, on the other hand, like part of packaging, does seem to be a considerable mystery, especially in light of the C-value “paradox”, here restated as the TAD-expansion paradox. This reviewer would appreciate some discussion of the hypothetical functions that might be provided by the balanced distribution of CNEs within TADs of different length, as if the evenness of the distribution was under selection without regard to total length; folding seems to fit this sort of phenomenon.

The C-value paradox indeed includes the expansion of TADs to approximately the same extent as the rest of the genome: we see a clear relationship between the scaling of GRB/TADs and genome size (Figure 5).

We fully agree with the reviewer that the folding must be preserved against the expansion of the genomic region, and that the equivalence of GRB and TAD spans we report here suggests that CNEs could play an as of yet undiscovered role in that folding. At the moment we do not have a testable hypothesis about the mechanism involved, even though we tried to investigate some of them - e.g. we and other groups tried to detect both local and global sequence complementarities that would allow CNEs to base-pair with each other, either directly or via RNA intermediates, but so far no such pervasive complementary sequences were found. The purpose of this paper is rather to provide all the evidence for GRB-TAD equivalence and to spur the interest in the community for finding the mechanism behind it.

We have now added a separate paragraph on this topic to the Discussion, as requested:

“Because of the unknown reason for the extreme conservation of CNEs, their function as enhancers, and the fact that their distribution closely follows the span of the TADs around genes known to be involved in long-range developmental regulation, it is tempting to speculate that CNEs are somehow directly involved in the chromatin folding of TADs, precisely arranging promoters and long-range enhancers in 3D space. Indeed, just like TADs, the spatial proximity of promoters and developmental enhancers seems to be stable across different cell types, regardless of the activity of either. At present there is no evidence from sequence analysis that CNEs are involved in sequence-mediated interactions, and their role in chromatin folding remains an open question.”

The METHODS seem to be state of the art.

Minor. Figure 4 legend, l4. Perhaps replace “...against repeat element insertion.” with “... against the retention of repeat element insertion.”

Fixed.

Reviewer #2:

Harmston et al. have conducted a genome-wide research where they try and assess the correspondence between TADs and evolutionarily conserved regulatory landscapes. To this aim, they first identify conserved non-coding elements (CNEs) present in various vertebrate genomes and displaying different degrees of synteny with the human genome. Subsequently, they group these CNEs into genomic regulatory blocks (GRBs) following different methods. They then analyse the distribution of these GRBs in the context of topologically associating domains (active or inactive) as well as with other elements like the presence of CTCF. The conclusions of this study generally support those of previous works, i.e. that TADs often include regulatory landscapes with a number of enhancers and their target genes, and that TADs are usually kept throughout evolution. These results are in many cases somehow redundant and it was not always easy to us to see the added value of this particular piece of work. Although the initial hypothesis is well accomplished, there is no challenge of previous models or, alternatively, a proper experimental approach to either confirm or reject the

presented results. Therefore, we support publication of this paper with a moderate enthusiasm only and would rather see it appearing in a more 'technically oriented' forum.

While the question to know whether or not the presence of CNEs is in a way reflected by the existence and/or structure of TADs is valid, it is now well established that dense regulatory landscapes –which contains many CNEs- do generally coincide with TADs structures (the question of causality is however still unclear). Therefore, we are not fully convinced by the novelty of this study, when taken at a general level. In addition, there are some technical concerns about how the results were generated and how their graphic and statistical representations are formalized, as well as some conceptual problems, to our opinion. Some of these issues are listed below.

Regarding novelty, the main novel insight of this work is the close correspondence between TADs and chromosomal regions spanned by clusters of extremely conserved non-coding elements. While it has been suggested previously that TADs correspond to regulatory domains (Dixon et al. 2012; Symmons et al. 2014) we report two contributions to the understanding of TAD function and evolution in manuscript that are neither already known nor do they follow straightforwardly from the present knowledge in the field:

1). Given the correspondence between CNE-rich regions and the strongest, most stable TADs; we propose that not all TADs represent the same kind of regulatory domain. TADs enriched for CNEs coincide with the span of tight clusters of CNEs and represent the regulatory domain of the trans-dev gene located within it, and while it may contain other genes and their regulatory elements its primary purpose is to ensure proper regulation of this gene with role in the regulation of multicellular processes, previously defined as GRB target gene (first defined in (Kikuta et al. 2007) reviewed in (Harmston and Lenhard 2013).

2). The correspondence between TADs and GRBs in different metazoan phyla suggests that this type of organisation has been under intense selective pressure, and acting on the same functional subset of genes, at least back to the last common ancestor of tetrapods and arthropods 800Mya ago.

• The funnel shaped density maps are used extensively. Should this article be targeted to a wide readership, some more information should be added either on the axis or in the figure legend in order to make understanding easier. Also, the work lacks proper statistical analysis: we do not think it is appropriate to talk about concordance simply because two plots have similar shapes.

Thank you for the suggestion - we have added informative titles to all of the funnel plots presented in the manuscript to help the reader understand what is being shown.

The idea behind the funnel-shaped heatmaps is straightforward - one feature is sorted by size and centred, giving the funnel shape. Then an independent feature is shown for the same regions, and if it reproduces the funnel shape, it means that the two features highly coincide in the genome. If the independent visualised feature is not spatially correlated with the original feature, it will not show the funnel shape.

To quantitatively assess the extent to which GRB and TADs coincide, we have added statistics on the distances between TAD and GRB boundaries that are now described briefly in the text, Figure S3 and Supplemental Table 1. We now report that 235 GRBs that lie within 120kb of a TAD boundary (H1

ESCs). We have calculated the same statistic for all of the comparisons investigated and have calculated p-values to assess the significance of identifying a specified number of GRBs having both boundaries located within Xkb of a TAD boundary. This was accomplished by randomly permuting the locations of GRBs (see Methods); after excluding centromeric regions. The number of GRBs found to have both edges located within a specific distance (120kb for human/40kb for *Drosophila*) of the nearest TAD boundary was found to be significant ($p < 1e-5$). Therefore the distribution of CNEs predicts the location of a subset of TAD boundaries with a degree of accuracy higher than that expected by chance.

• The authors like to emphasize the good correspondence between TADs and GRBs. TADs are chromatin regions originally defined by their structural interaction profiles. Previous studies have already correlated regulatory landscapes (including enhancer-promoter interactions) with the distribution of TADs. In their approach, the authors make the assumption that conserved regions display presumed regulatory potentials. This referee has some difficulty to see where the novelty stands in this proposal, with respect to previously published papers (including reviews).

Previous studies have correlated regulatory landscapes with the distribution of TADs (e.g. Dixon et al. 2012; Symmons et al. 2014) however a number of these studies suggest that TADs can function as a co-regulatory environment, where the TAD represents the regulatory landscape for all genes located within the TAD. The strong overlap between GRBs/CNEs and a subset of TADs, as shown in this manuscript, argues against this model of TAD function. In a GRB-TAD, while there can be several genes, the regulatory landscape is typically relevant to one (the GRB target gene), in contrast to these models of TAD function. Previous work on CNEs and GRBs has found that CNEs regulate one specific gene which function in development and morphogenesis (or, in rare cases, on ancient paralogous clusters of developmental genes such as HOX or IRX genes), and do not appear to have a significant effects on nearby housekeeping genes also spanned by the GRB. While these TADs do contain regulatory elements for housekeeping genes located within them, our results suggest that as genomic regions they primarily represent the span of the regulatory domain of the key developmentally regulated gene located within the GRB. There is a large body of work that has found that CNEs act as regulatory elements in reporter constructs (e.g. Vista Enhancer Browser, <https://enhancer.lbl.gov>, is a database dedicated to testing enhancer activity of individual human CNEs in mouse embryogenesis). One of the findings of our manuscript is that the distribution of these conserved regions with regulatory potential corresponds in a number of cases to the structural/topological organisation of the genome.

• If the aim of this work is to compile conserved regions with regulatory potential, several epigenetic marks (e.g. H3K4me1) could have been used instead of identifying such regions, as well as to remove promoters that would complicate the interpretation. In fact, the authors address this question of whether or not epigenetic marks could be a valuable parameter to identify GRBs. However, the use of H3K27ac datasets from different cell lines is likely not suitable for this purpose, since many genes and/or their enhancers may not be active in the cell lines analysed. To overcome this problem, the authors may advantageously use other parameters, which could be taken as hallmarks of poised enhancers like p300 or K4me1 or ATAC-seq datasets. In this case, enrichment should be detected even in the absence of transcription.

We have followed the reviewer's suggestion and generated an additional series of plots utilising H3K4me1 ChIP-seq data. As can be observed in Figure S4C, there is no correspondence between the distribution of H3K4me1 and the boundaries of these regions - H3K4me1 signal fails to reproduce the funnel shape of GRBs. We attribute this result to the fact that while the previous evidence suggests CNEs and the GRBs they define are dedicated to long-range developmental regulation, only a subset of H3K4me1-marked elements are involved in this type of regulation in a specific context. As

a result we do not expect the distribution of H3K4me1 to sharply distinguish between GRBs (and, as reported in this manuscript, their coincident TADs) and the neighbouring regions.

•The authors rightly argue that regions positive for H3K27ac do not allow on their own to define GRB with accuracy, because they are also enriched in promoter regions. This limitation could be fixed by removing in silico either all TSS surrounding regions, or all loci covered by H3K4me3.

We have now replotted the relevant heatmaps after removing H3K27ac signal that overlaps Ensembl-defined TSSs (± 2.5 kb). As shown in the updated Figure S4B, we still cannot observe any obvious pattern demarcating these regulatory domains. At some boundaries there appears to be an increase in regions devoid of signal, however this is due to the presence of housekeeping genes near the boundaries. These results confirm that H3K27ac marks cannot distinguish GRBs, or the coinciding TADs, from their surroundings, even after removing the H3K27ac peaks that overlap promoters.

• While a big proportion of GRB overlaps with a single TAD, an important proportion also encompasses more than one, sometimes even more than two TADs. It is not clear to this referee as to why a definition of GRB based on synteny and density of HNCs would be a better predictor of GRBs than the TADs themselves. Also the majority (if not all) of the examples provided in this study correspond to developmental genes flanked by relatively large gene deserts/regulatory landscapes. It is thus not entirely clear whether this approach would also identify GRB in more gene-dense regions or, alternatively, in TADs enriched in housekeeping/non-developmental genes?

TADs are defined by folding data, while GRBs are defined by CNEs: this manuscript describes how these two concepts converge and that at a subset of loci they represent the same underlying structure. At the majority of loci TAD boundaries would determine GRB boundaries more precisely than the density of CNEs - as commented in the discussion. We have added an additional analysis of Hoxd 4C-seq data in developing limb and forebrain (Figure S7), which highlights a situation in which the span of CNEs reflects the regulatory domain better than the TAD structure as at this regulatory domain corresponds to two TADs.

In addition, there are a number of technical and biological limitations associated with identification of both TADs and GRBs. TADs reflect a segmentation of the genome from a complex hierarchical 3D structure into a simple set of intervals. The algorithms used for performing this segmentation have issues with different scenarios. In a number of cases, HOMER calls boundaries at the promoters of genes where there is a small change in directionality index, whereas in some cases DIXON appears to miss a subset of TAD boundaries that are apparent in the interaction matrix. Some of the examples of a called GRB covering two called TADs is a consequence of these technical issues with TAD callers, but in other cases this is due to not being able to separate closely spaced CNEs into distinct blocks (see Figure 3C), discussed on page 6 in the manuscript. In addition, defining the boundaries of TADs is highly dependent on the resolution of the Hi-C experiment, so our ability to detect concordance between TADs and GRBs is automatically limited by this. We believe that TADs and GRBs can provide complementary sources of information about putative regulatory domains.

CNEs are preferentially found around developmental genes, which are often associated with large gene deserts. Therefore, TADs that are associated with CNEs are associated with lower gene density (Figure 4F). In general, CNEs are not found in gene-dense regions or regions containing lots of constitutively active genes: these genes do not tend to be under complex spatio-temporal regulation by long-range elements (reviewed in (Harmston and Lenhard 2013)). However, we do identify a number of TADs which contain a developmental TF but are not surrounded by a gene desert - instead, most of the GRB is spanned by target and bystander genes, with most CNEs contained in

their introns. As an example, *ISL2* is a LIM homeobox located in a TAD (chr15:76470001-77230000) which is largely covered by genes, including a long, ubiquitously expressed gene with large and numerous introns (*SCAPER*). A number of CNEs are located within these introns and the microsyntenic relationship between these *ISL2* and *SCAPER* is conserved over large evolutionary distances (Irimia et al. 2012) We have added a description and brief discussion of this locus to Supplemental figure S6.

• An important fraction of GRBs correlate with TADs, the authors say. Interestingly, they show that 'GRB-TADs' tend to have a lower density of transposons and that they switch from activity compartment A to B and vice-versa more than non 'GRB-TADs'. Several studies have correlated TAD borders with a number of features including the density of CTCF binding sites. GRB-TADs also seem to be more enriched in constitutively bound CTCF at their borders than non GRB-TADs. These observations are interesting indeed, yet should the identification of GRB be biased in favour of identifying GRBs in regions that are well organized into TADs (see point 1), this would naturally impact upon the interpretation of the results.

We do not use any information from chromosome conformation capture experiments in generating GRB span/boundary predictions - these regions are estimated purely from the density of syntenic CNEs, i.e. from genomic sequence comparisons. In other words, we call GRBs first, and then compare them to TADs. As such we do not think there is any bias that would result from previously known correlations. The evidence we present here suggests that regions that have high levels of non-coding conservation are well organised into TADs, and that high density of non-coding conservation is a good sequence-derived predictor of the extent of those TADs.

• One conclusion of the paper is that GRBs are as good predictors as TADs of the existence of regulatory landscapes around given genes. Could the authors provide evidence that GRBs predict the presence of a regulatory domain as efficiently as Hi-C-defined TADs? Can GRBs be found also in inter-TAD regions? Examples would be nice to illustrate these two points.

In this manuscript, we do not make a strong statement regarding the ability of CNE density/GRBs to predict the boundaries of TADs as efficiently as Hi-C. Rather, we aim to propose that CNE density predicts the boundaries of a specific subset of TADs with a level of accuracy that suggests strong functional association. As shown in Figure 2C and S3A we do detect a number of GRB calls that do not overlap with TADs. In some situations these reflect regions of the genome where the TAD callers are unable to identify TADs robustly, e.g. there appears to be some issues with calling TADs proximal to telomeres and centromeres, although from the Hi-C interaction maps they appear that they should be identified as TADs.

• The authors fall short in estimating which structures predated the other. Did GRBs appear before TADs were established and organized? Or the contrary? Is the former the result of the latter? Was there a selective advantage to establish 'private' chromatin structures in order to prevent illegitimate enhancers-promoter contacts? Here the general lack of an experimental approach makes this sort of conclusions difficult, which seriously limits the impact of the study. Hi-C datasets (or 4C-seq datasets) on some of the compared species would definitely bring more power to the hypothesis.

At a set of loci GRBs and TADs directly correspond to the same regulatory domain; if they are two manifestations of the same regulatory phenomenon, then TADs did not arise before or after the GRB because they represent the same structure. Given results from work investigating the impact of

perturbing the boundaries of TADs (Lupiáñez et al. 2015) and the effect of translocations within the regulatory domains of developmental genes on their expression (Symmons et al. 2016) it seems that there is a strong selective advantage to establish “private” or “restricted” chromatin structures around trans-dev genes to prevent their ectopic regulation. We have added a sentence to explicitly state this in the discussion of the manuscript “*It is highly probable that this type of restricted topological structure arose early in Metazoan evolution as a result of strong selective pressure to prevent this.*”. We disagree that the lack of an experimental approach limits the impact of the study as CNEs, GRBs and TADs all have extensive experimental evidence associated with them, which is in line with the predictions of our GRB-TAD model.

Methodology

***For the identification of conserved non-coding elements, three different kinds of regions were excluded: exonic sequences, repetitive sequences as well as sequences present at least four times in the genome. Could this latter criterion exclude certain discrete binding sites for transcription factors?**

The criteria used to identify CNEs will exclude standalone conserved binding sites regardless of the subsequent filter for repetitive sequences and sequences present four or more times in the genome. CNEs do not correspond to discrete highly conserved TFBSs but are much longer. CNEs, identified using 70% over 50bp between human and chicken, are on average 175bp long, and range in length from 38bp (allowing gaps) to 2317bp (N=86757). TFBS motifs are much shorter (most are between 8bp and 20bp) and allow for multiple nucleotide substitutions. Clusters of multiple TFBSs are unlikely to occur in identical patterns in multiple locations, and so will not be affected by this criterion.

***Also, in the processing of Hi-C datasets, why were heterochromatic chromosomes discarded from further analysis? The authors should justify their choice in the text.**

Heterochromatic chromosomes were discarded due the difficulties in mapping Hi-C reads to these chromosomes, which resulted in extremely low levels of uniquely mapped reads. Discarding of reads mapping to heterochromatic chromosomes/ignoring them completely is often done in analysis of *D. melanogaster* Hi-C datasets (i.e. Sexton et al. 2012). We have added a clarification to the Methods section.

***In their calculation of the expected number of GRB boundaries lying within a specific distance of a TAD boundary, the authors consider that if both GRB borders are within three Hi-C bins of a TAD border, they are taken as ‘coincidental’. If the Hi-C data were binned at 20kb, this means that the ‘coincidental region’ is around 60kb (or 120kb if the window size is what the authors seem to take as a reference -see on line 602). However, looking at Fig. 2D, it seems that 50% of the GRBs analysed display borders at a distance of 250kb from a TAD border. Taking into account that TADs are 800kb large on average in mammals, 250kb would represent more than a quarter of their length, while such difference is not appreciable in the heatmap graphs (Fig. 2A). Also, this would mean that a large proportion (around 50%) of GRB borders would not coincide with the TAD border according with the criteria used by the authors. Could the authors provide an estimation of the average distance between GRB border and TAD borders? Alternatively the authors could provide a summary table with the numbers of GRBs and TADs with and without overlapping borders and discuss this discordance?**

As shown in Figure 2A, the funnel shape indicating concordance between GRB and TAD spans is lost at the very bottom of the plot. We believe that some of these GRBs correspond to GRBs that were

split into two or more putative regions when they should not have been (we have added a brief description of this to the Methods section). This may occur if there is a sparsely populated region of CNEs between two dense regions, or if the distance between two putative regions is large. This factor accounts for a large proportion of the GRBs which have both borders at a distance >250kb from the nearest TAD boundary.

The distribution of relative distances from GRB borders to the nearest TAD boundary is shown in Supplemental Figure S3 E and F. As suggested by the reviewer, we have added statistics on the average distances (both relative and absolute) between GRB orders and TAD borders to Supplemental Table S1.

In addition, as discussed above, we have performed permutation testing on the number of GRBs which have both edges located with Xkb of a TAD boundary and found that this correspondence across all of the situations studied in this manuscript is significant ($p < 1e-5$) - see Figure S3G and corresponding figure legend.

***The classification GRB-TADs versus non-GRB-TADs is not easy to understand. A little scheme with a graphical depiction of the classification could be advantageously added as supplementary material. Also, there seems to be a mistake on line 545 in the description of non-GRB-TADs: 'TADs with less than a 20% overlap with a TAD were assigned as non-GRB-TADs...'. And how much overlap between two TADs was considered in order to discard them from the analysis? And consequently, how many TADs were classified as belonging to either one or the other category?**

Thank you for spotting the mistake on line 545; this has now been corrected. We have added a schematic (Figure S8A) illustrating the fraction of overlap between GRBs and TADs, which shows a clear bimodal distribution that we used to select the thresholds. In addition we have provided a new supplemental table (Table S1) that contains statistics on numbers and sizes of CNEs, GRBs and TADs, along with numbers corresponding to GRB-TAD/nonGRB-TAD classifications.

***Regarding repetitive elements, why were only SINEs used in the analysis?**

We presented results from our analysis of SINE elements primarily due to the observation reported in Dixon et al. that SINE density was a good predictor of TAD boundaries (Dixon et al. 2012) Our analysis suggests that this enrichment of SINE elements at the boundaries of TADs is an artefact generated by averaging over TADs which are depleted in transposons and TADs which are not. The reviewer is correct that Simons et al. reported a presence of transposon-free regions as regions larger than 10kb from which SINEs, LINEs, LTRs and DNA transposons are absent (Simons et al. 2006) We have included our analysis for these additional classes of transposons. GRBs and GRB-TADs do not appear to be depleted or enriched for DNA transposons compared to other regions, whereas there appears to be a slight depletion in the levels of LINEs and LTRs in these regions although not as striking as the differences in SINE density. In general there appears to be a lower levels of LINEs at the boundaries of TADs, regardless of whether they contain CNEs or not, but this may be a secondary consequence of the fact that LINEs and boundary elements are mutually exclusive. We have added the results of these analyses as additional panels to Figure S8.

***Regarding CTCF binding at GRBs and TADs, why was a width of 400bp established for the consensus peaks?**

The average size of CTCF peaks across all of the datasets investigated was 404bp (median 360bp). Hence we used 400bp as the width of our consensus peaks. We have added an explanation of this to the Methods section.

***The distance selected to define the boundary region (10kb) is also unclear. It seems that GRBs are often shorter than TADs (see comment above). Thus, the distance between GRB borders and TAD borders should be taken into consideration in order not to under-evaluate the relationship of CTCF with GRB borders. This may indeed have an impact upon the interpretation: if a given TAD evolved by containing important enhancers for developmental genes, some presumably constitutive architectural contacts may have evolved to limit the extension of enhancer activity to prevent aberrant expression of surrounding genes. In this context, TAD border may coincide closely to GRB borders and this section of the paper is not fully clear.**

Since all CNEs of a GRB are expected to be inside the TAD that coincides with the GRB, the estimate based on CNE density is expected to be slightly shorter than the TAD span, depending on the distance between the outermost CNEs and the physical TAD boundary (see Supplemental Figure S3E/F). This also explains the one-sided skew between CTCF signal and the estimated GRB boundary (Figure S9A) - the GRB-based boundary estimate is expected to be close to the TAD boundary, but always on the inside of it. We have tried to explain this better in the manuscript.

In order to confirm that our analyses are not biased by the use of a 10kb window around the borders of GRBs, we have repeated this analysis using a variety of distance thresholds. At all of the distances investigated (10kb-120kb in step sizes of 10kb) the observed enrichment for constitutive CTCF binding and depletion for cell-type specific patterns of CTCF binding was maintained (see below). We have added a comment to the Methods section of the paper to state that we have confirmed that these enrichments are stable across distances.

Distance	Specific	Intermediate	Constitutive
10kb	1.0373330e-08	4.443693e-02	3.990424e-11
20kb	2.059616e-10	2.021779e-03	2.319501e-09
30kb	1.906641e-11	1.965702e-03	3.249330e-11
40kb	5.165246e-09	2.937340e-02	1.131721e-11
50kb	8.531262e-10	2.297566e-02	6.520968e-13
60kb	1.273475e-09	1.058567e-02	4.646904e-11
70kb	5.390368e-08	3.025290e-02	7.831619e-10
80kb	6.640935e-07	5.385307e-02	8.495329e-09
90kb	1.750346e-06	6.140217e-02	2.949270e-08
100kb	5.390183e-07	1.997877e-02	3.246044e-07

110kb	4.097632e-07	3.061997e-02	3.665846e-08
120kb	4.035278e-07	9.628084e-03	2.257332e-06

Minor comments

Line 107-114. Human versus Gallus. 819 GRBs when using 70% identity of 50bp CNEs. Human vs Gar. 719 when using 70% identity of 30bps CNEs. Why were different lengths of CNEs considered for each species? Is it because identifying CNEs becomes more difficult at longer evolutionary distance? This should be explained in the text.

This is correct. Despite extreme conservation, at larger evolutionary distances CNEs do accumulate more mutations so lower thresholds for size and level of conservation are used to extract a comparable set of CNEs (see (Engström et al. 2008)). We have added a comment on this in the methods section of the manuscript.

Fig1A. The figure is divided into different panels showing: chromosome position in the top, CNEs density distribution and a smooth version of the same. Also, several bars representing the definite borders of these GRBs are shown. Although, it seems relevant to show the different degrees of concordance according to the level of assumed identity, the plotted information is redundant. The same results are plotted in almost two identical ways (discrete densities and smoothed densities).

In Figure 1A, the first five panels show individual CNEs identified at various thresholds, rather than showing a discrete density distribution. While in some ways the distribution of discrete CNEs and smoothed densities is partially redundant, we think that it is useful to include both. The method we have used to identify CNEs uses a thresholded sliding window approach and so CNEs can exhibit some overlap between each other. Readers of the manuscript may not have examined the distribution of CNEs before and this should make it easier to visualise how they are distributed throughout the region and what the distribution of the discrete CNEs underlying the smoothed densities look like.

Fig1B. The authors plot the distribution of all analysed human versus Gallus GRBs according to their lengths. It would be useful to better understand what is the average size of the GRBs. If the authors want to state that approximately 80% of the conserved GRBs are shorter than 3Mb in length, they should include this in the text.

We have now generated a supplemental table (Supplemental Table S1) that contains summary statistics on the size of the sets of putative GRBs for all comparisons investigated. The median GRB size for hg19/galGal4 (70%/50bp) is approximately 880kb.

Fig1D. This graph is generally difficult to understand. What is exactly plotted there? Were the different GRBs in each species first centred, ordered and then overlapped? Which set of hg19-

galGal4 (70%-80%-90%) was used? It seems that some explanations come later in the text (line137-140), but this should be stated when introducing the figure.

We have updated the figure legend for Figure 1D to provide a clearer description of the figure. In this figure GRBs called between human and chicken using 70%/50bp were used. Then GRBs identified using CNEs identified in hg19-monDom5 and hg19-lepOcu1 comparisons were overlaid to show the large scale concordance between the GRBs identified using different set of CNEs.

In line 121, it is stated that the subset of conserved GRB boundaries include developmental regulators. Only MEIS1 and IRX3 are used as examples. A comprehensive list of these regulators would give more power to this affirmation.

The target genes of GRBs are typically developmental transcription factors or cell adhesion proteins. These genes are typically members of PAX, HOX, FOX, SOX and other DNA-binding families. We have mentioned in the text the presence of conserved boundaries at other key developmental TFs i.e. *PBX1*, *OTX1*, *LMO4* and *PROX1* etc. We have also added text pointing the reader towards Akalin et al. which has a supplemental table listing a curated set of putative target genes (Akalin et al. 2009) The set is reliable but not complete: we are working on a method for the automatic identification of target genes for each GRB.

Figure2. It would be interesting to see an overlay between the different dataset (GRB and Hi-C directionality index). By superimposing the two distribution heatmaps, the concordances and differences at the edge of GRBs would be better spotted.

Please find attached a figure showing the overlay of putative GRB boundaries with Hi-C directionality index. As we have reported the figure shows a strong concordance between changes in Hi-C directionality and the boundaries of putative GRBs.

Fig2A. Are the Hi-C data-sets good enough to properly map them at 5kb resolution? And why is the 5kb bin distribution added to the figure? Is there any extra information there?

The datasets are not of sufficient quality/depth to be able to examine interactions at high resolution. The Hi-C data-sets were mapped using a bin size of 20kb (see Methods). Only for visualisation was a 5kb window used. We have generated a series of plots using different bin sizes (10kb, 20kb; see the file *Figures_for_reviewers.zip* in the submission) to show that the observed pattern is not dependent on the choice of bin size.

Fig2C. It is well-established that TADs do not dramatically change between cell lines. Why then was the GRBs-TAD correlation supposed to change?

While we did not expect the GRB-TAD correlation to change between cell lines, we felt it prudent to make this statement and confirm the applicability of our findings to multiple cell lines. As expected from previously published previous analyses, our analysis did not reveal any major differences in the organisation of TADs between the datasets investigated. However, while TAD structures are largely invariant between cell lines and tissues, there is some amount of variation across cell lines and tissues (Schmitt et al. 2016) and additionally technical issues can have major effects on boundary estimation. We wished to show that the correspondence between CNE density and TAD structure is robust over a set of datasets corresponding to developmentally relevant lineages/stages.

Several examples of GRBs that fit TAD distribution are given. However, two of these examples seem to point to the opposite direction. In line 198-201 and Fig3, the region encompassing TOX3 and SALL1 belongs to the same GRB, whereas the interaction matrixes clearly show a split distribution of contacts. In Fig S5, something similar occurs at the HoxD locus. How do these examples integrate the general conclusions?

We added the example in Figure 3 to illustrate a number of points. First the strong correspondence between the *IRX3/5/6* TAD and CNE density, but also to highlight that CNE density alone cannot provide adequate information to separate very closely spaced GRBs in some cases. Enhancer trap studies using CNEs have demonstrated that there are two separate regulatory domains, with one encompassing *TOX3* and one containing *SALL1* (Pennacchio et al. 2006) This separation is apparent in the Hi-C data but it is not possible to separate these domains based on CNE density alone. We were attempting to highlight that while on the whole our analysis finds that CNE density can be a good proxy for inferring regulatory domains (i.e. *IRX3* TAD) but it is not perfect (*SALL1/TOX3* TAD). This is a case where GRB and TAD data complement each other - GRB to indicate that this is an area where long-range developmental regulation occurs, TADs to provide precise boundaries, and GRB model again to suggest the likely target gene under long-range regulation.

The HoxD locus appears to show strong evidence of hierarchy within both its centromeric and telomeric TAD. Depending on the developmental context, different genes within the HoxD cluster interact with regulatory elements located either in the telomeric or centromeric TAD. The proposed HoxD GRB encompasses both of these TADs (Figure S6A) and a similar pattern is observed at this locus in mouse (Figure S7A). Regulatory information located within both TADs is required for proper limb development (Montavon et al. 2011) with the regulatory landscape of this locus better predicted by synteny than by topological structure.

Fig S6. There may be a risk to try to extrapolate genome wide a situation observed only at one locus.

We completely agree with the reviewer on the risk of extrapolating from a single locus to a genome-wide statement. In the text, we have specifically stated that this observation is from one locus only and that this provides *limited evidence* for the correspondence between CNE density and span of regulatory interactions in sea urchin. In addition to the genome-wide patterns observed in *Drosophila* and human, this correspondence is supportive of our main hypotheses and findings.

The sentence: 'changes in SINE density at TAD boundaries is indicative of transitions between TADs under high negative selection against insertions and TADs which are not under this form of selective pressure' is unclear to us. Why is it not a local effect that can affect only boundary regions?

We do not believe that this is a local effect, as we observe a depletion in SINE density throughout TADs associated with GRBs, rather than a depletion specific to boundary regions (Figures 4A and S8B). We agree that SINE elements can impact genome organisation, as previously observed (Lunyak et al. 2007; Schmidt et al. 2012) However, the depletion of SINE insertions suggests that the cis-regulatory architecture of these regions is generally unable to tolerate them, potentially because they can alter CTCF binding or contain proto-binding sites for TFs/CTCF (Bourque et al. 2008)

Line 671. Typo: Loess-smoothed.

Fixed

Reviewer #3:

Harmston et al. present evidence that highly conserved noncoding sequences, or CNEs, are often located at the boundaries of topological association domains (TADs) containing developmental control genes. This is an important observation and certainly worthy of publication in Nat Comm. I have just a few minor comments for the authors to consider prior to publication:

1. It would be helpful for the authors to explicitly state what % of the ~3000 TADs in the human genome contain CNEs/GRBs at their boundaries. How do these number change with a sliding definition of CNEs and GRBs?

We have now stated within the main text the number of TADs that we classify as been largely covered by CNEs. We have generated a supplemental table (Table S1) that contains information on the number of TADs identified as GRB-TADs or non-GRB TADs at various CNE thresholds. In addition, we have added a new panel (Figure S8A) that demonstrates how the choice of overlap threshold was determined. Approximately 20% of TADs are classified as GRB-TADs using our approach. In addition we have provided more in depth statistics on the correspondence of boundaries between GRBs and TADs in the manuscript and Table S1.

2. Do the particular human TADs containing GRBs coincide with regions of "deep synteny" described by Putnam et al. Nature 2008?

In short, no. Putnam et al. found widespread evidence of macro-synteny between vertebrates and amphioxus, where clusters of genes on one contiguous part of chromosome in one species are also present on one contiguous part of one chromosome of another - irrespective of whether the order of genes is conserved or if other genes are interspersed in the same region in one species but not the other. The regions reported in Putnam et al. (Supplemental Table 14) are much larger than the regions detailed in this manuscript. In this manuscript we use a genomic sequence definition of synteny: a collinear arrangement of genes and other conserved elements (predominantly CNEs) across the two compared genomes. This is a standard way of defining synteny in the context of long-range regulation. This level of synteny is rare between human and *Amphioxus*. Confusingly, the term *microsynteny* is sometimes used to denote short instances of our definition of synteny between evolutionarily distant genomes, such as those between *Drosophila* and mosquitoes. To avoid misunderstanding, we have now explicitly defined what we mean by synteny in this manuscript.

Although the available space does not permit a detailed treatment, we were intrigued by this question and decided to check it for future work. While maintenance of synteny (as collinear arrangement of genes and noncoding elements) between human and *Amphioxus* is extremely rare, Irimia et al. have reported some conservation of microsynteny (Irimia et al. 2012) We have performed an initial investigation of reported set of conserved gene pairs present between human and *Amphioxus* and the topological domains identified in this manuscript. Of the 735 reported *human-Amphioxus* pairs with valid gene identifiers in Ensembl release 79 whose promoters did not overlap (\pm 2.5kb), 663 pairs (96%) were found to be located in the same TAD in at least one of the sets of TADs investigated (across lineages/methods). These pairs of genes contain pairs whose microsynteny is maintained due to cis-regulatory constraints (i.e. one member contains regulatory information for the other - i.e. *MICU2/FGF9*, *SUPT3H/RUNX2*) or which are under a different set of constraints due to their co-

regulation (i.e. histone clusters, SERPINs, cytochrome P450). However, it is not possible to reliably filter this list of pairs to only contain those that are microsyntenic due to cis-regulatory constraints. It appears that 176 (24%) of these pairs are located within regions overlapping GRBs. 111 (15%) of these pairs overlap TADs that are classified as GRB-TADs in at least one of the sets of TADs investigated. However, examination of the list of microsyntenic pairs revealed that a large number of them correspond to paralogous expansions of gene sets not necessarily under long-range regulation, whereas the maintenance of microsynteny due to cis-regulatory constraints typically involves structurally and functionally unrelated genes (such as GRB target and bystander genes).

3. It might be useful to include a more in-depth analysis of the Hox clusters (e.g., Hoxd) since these include a number of notorious TADs that have been defined in functional terms.

We agree that the Hox clusters are interesting loci to examine using this approach. Interestingly previous reports have reported that the extent of regulatory elements controlling the HoxD locus is better approximated by synteny and CNE density rather than Hi-C. In order to address this question, we have now performed a reanalysis of publicly available 4C data specifically investigating the interaction patterns of *HoxD4* and *HoxD13* in the mouse limb and brains, specifically investigating the involvement of CNEs, see Figure S7A. Montavon et al. investigated the interaction landscape of *HoxD13* and *HoxD4* in brain and limb using 4C in mouse (Montavon et al. 2011) and identified that *HoxD13* appeared to largely interact with elements present in the centromeric TAD whereas *HoxD4* appeared to largely interact with elements located in the telomeric TAD. In the developing limb we find that *Hoxd13* interacts with 28 CNEs located within the centromeric TAD and 11 CNEs located within the telomeric TAD, whereas in embryonic forebrain *Hoxd13* interacts with 26 CNEs located with the centromeric TAD and 15 CNEs within the telomeric TAD. In the developing limb *Hoxd4* interacts with 7 CNEs located with the centromeric TAD and 39 CNEs within the telomeric TAD, whereas in embryonic forebrain *Hoxd4* interacts with 23 and 35 CNEs located with the centromeric and telomeric TADs respectively. These results highlight the involvement of CNEs in the regulation of *HoxD* cluster. In mouse embryonic limb, *Hoxd13* is preferentially interacting with elements in the centromeric TAD and *HoxD4* preferentially interacting with elements in the telomeric TAD. However, CNEs appear to be involved in interactions with HoxD in both embryonic limb and brain, although HoxD does not show expression in brain. This suggests that at specific stages the spatial proximity of CNEs with their target promoters within TADs is present regardless of the activity of either (Ghavi-Helm et al. 2014) We have described the results of this analysis in the figure legend for Figure S7.

References

- Akalin A, Fredman D, Arner E, Dong X, Bryne JC, Suzuki H, Daub CO, Hayashizaki Y, Lenhard B. 2009. Transcriptional features of genomic regulatory blocks. *Genome Biol* **10**: R38.
- Bejerano G, Pheasant M, Makunin I, Stephen S, Kent WJ, Mattick JS, Haussler D. 2004. Ultraconserved elements in the human genome. *Science* **304**: 1321–1325.
- Bourque G, Leong B, Vega VB, Chen X, Lee YL, Srinivasan KG, Chew J-L, Ruan Y, Wei C-L, Ng HH, et al. 2008. Evolution of the mammalian transcription factor binding repertoire via transposable elements. *Genome Res* **18**: 1752–1762.

- Dixon JR, Selvaraj S, Yue F, Kim A, Li Y, Shen Y, Hu M, Liu JS, Ren B. 2012. Topological domains in mammalian genomes identified by analysis of chromatin interactions. *Nature* **485**: 376–380.
- Engström PG, Fredman D, Lenhard B. 2008. Ancora: a web resource for exploring highly conserved noncoding elements and their association with developmental regulatory genes. *Genome Biol* **9**: R34.
- Ghavi-Helm Y, Klein FA, Pakozdi T, Ciglar L, Noordermeer D, Huber W, Furlong EEM. 2014. Enhancer loops appear stable during development and are associated with paused polymerase. *Nature* **512**: 96–100.
- Harmston N, Baresic A, Lenhard B. 2013. **The mystery of extreme non-coding conservation.** *Philos Trans R Soc Lond, B, Biol Sci* **368**.
- Harmston N, Lenhard B. 2013. Chromatin and epigenetic features of long-range gene regulation. *Nucleic Acids Res* **41**: 7185–7199.
- Irimia M, Tena JJ, Alexis MS, Fernandez-Miñan A, Maeso I, Bogdanovic O, la Calle Mustienes de E, Roy SW, Gómez-Skarmeta JL, Fraser HB. 2012. Extensive conservation of ancient microsynteny across metazoans due to cis-regulatory constraints. *Genome Res* **22**: 2356–2367.
- Kikuta H, Laplante M, Navratilova P, Komisarczuk AZ, Engström PG, Fredman D, Akalin A, Caccamo M, Sealy I, Howe K, et al. 2007. Genomic regulatory blocks encompass multiple neighboring genes and maintain conserved synteny in vertebrates. *Genome Res* **17**: 545–555.
- Lee AP, Koh EGL, Tay A, Brenner S, Venkatesh B. 2006. Highly conserved syntenic blocks at the vertebrate Hox loci and conserved regulatory elements within and outside Hox gene clusters. *103*: 6994–6999.
- Lunyak VV, Prefontaine GG, Nunez E, Cramer T, Ju B-G, Ohgi KA, Hutt K, Roy R, García-Díaz A, Zhu X, et al. 2007. Developmentally regulated activation of a SINE B2 repeat as a domain boundary in organogenesis. *Science* **317**: 248–251.
- Lupiáñez DG, Kraft K, Heinrich V, Krawitz P, Brancati F, Klopocki E, Horn D, Kayserili H, Opitz JM, Laxova R, et al. 2015. Disruptions of topological chromatin domains cause pathogenic rewiring of gene-enhancer interactions. *Cell* **161**: 1012–1025.
- Montavon T, Soshnikova N, Mascrez B, Joye E, Thevenet L, Splinter E, de Laat W, Spitz F, Duboule D. 2011. A regulatory archipelago controls Hox genes transcription in digits. *Cell* **147**: 1132–1145.
- Pennacchio LA, Ahituv N, Moses AM, Prabhakar S, Nobrega MA, Shoukry M, Minovitsky S, Dubchak I, Holt A, Lewis KD, et al. 2006. In vivo enhancer analysis of human conserved non-coding sequences. *Nature* **444**: 499–502.
- Schmidt D, Schwalie PC, Wilson MD, Ballester B, Gonçalves A, Kutter C, Brown GD, Marshall A, Flicek P, Odom DT. 2012. Waves of retrotransposon expansion remodel genome organization and CTCF binding in multiple mammalian lineages. *Cell* **148**: 335–348.
- Schmitt AD, Hu M, Jung I, Xu Z, Qiu Y, Tan CL, Li Y, Lin S, Lin Y, Barr CL, et al. 2016. A Compendium of Chromatin Contact Maps Reveals Spatially Active Regions in the Human Genome. *Cell Rep* **17**: 2042–2059.
- Simons C, Pheasant M, Makunin IV, Mattick JS. 2006. Transposon-free regions in mammalian

genomes. *Genome Res* **16**: 164–172.

Symmons O, Pan L, Remeseiro S, Aktas T, Klein F, Huber W, Spitz F. 2016. The Shh Topological Domain Facilitates the Action of Remote Enhancers by Reducing the Effects of Genomic Distances. *Dev Cell*.

Symmons O, Uslu VV, Tsujimura T, Ruf S, Nassari S, Schwarzer W, Etwiller L, Spitz F. 2014. Functional and topological characteristics of mammalian regulatory domains. *Genome Res*.

Wasserman WW, Sandelin A. 2004. Applied bioinformatics for the identification of regulatory elements. *Nat Rev Genet* **5**: 276–287.

REVIEWERS' COMMENTS:

Reviewer #1 (Remarks to the Author):

The authors have responded intelligently and adequately to their reviewers. Well done.

Reviewer #2 (Remarks to the Author):

I have been through both the responses of the authors and the modified manuscript and, while I am still not fully convinced by all their responses, I acknowledge their revisions and I would not oppose to publication. The central point is of interest and will initiate some discussion in the community.

A last comment: It has been proposed recently that in many instances, ancestral chromatin domains (TADs) may have triggered the formation of 'GRBs' (pleiotropic regulations) (Lonfat et al. 2014). Should this be the case, a 'correspondance' would be merely causal- Anything in these datasets that may address this issue?

Reviewer #3 (Remarks to the Author):

I am satisfied with the revised manuscript and support publication without further delay.

Response to reviewers

Reviewer #2:

I have been through both the responses of the authors and the modified manuscript and, while I am still not fully convinced by all their responses, I acknowledge their revisions and I would not oppose to publication. The central point is of interest and will initiate some discussion in the community.

A last comment: It has been proposed recently that in many instances, ancestral chromatin domains (TADs) may have triggered the formation of 'GRBs' (pleiotropic regulations) (Lonfat et al. 2014). Should this be the case, a 'correspondance' would be merely causal- Anything in these datasets that may address this issue?

Lonfat *et al.* observed that the regulatory domains around the HoxD and HoxA cluster are similar in size and organisation and commented onto potential mechanisms that could lead to this convergence, either i) “*presence in an ancestral Hox cluster of a constrained functionality or structure, which was used as a starting point to facilitate this convergence*” or ii) “*convergent enhancer evolution may have driven the emergence of comparable TADs at both loci*”¹. The correspondence between TADs and clusters of CNEs at orthologous loci in both *Drosophila* and human favours the first mechanism (as favoured by Lonfat *et al.*) and suggests that genome-wide most of the TADs located around developmental genes are ancestral. The presence of a single regulatory domain around the only Hox cluster in *Amphioxus*, a chordate that didn't undergo two rounds of whole genome duplication, also supports this hypothesis/mechanism². However, the conservation of this form of regulatory domain still does not explain the need for such high levels of non-coding conservation. We have added a comment referring to this contribution of our study to resolving this question posed by Lonfat et al, which now reads:

“This correspondence suggests that the regulatory domains around development genes are ancestral and not the result of convergent evolution⁵⁴, and may have existed since the origin of Metazoa.”

1. Lonfat, N., Montavon, T., Darbellay, F., Gitto, S. & Duboule, D. Convergent evolution of complex regulatory landscapes and pleiotropy at Hox loci. *Science* **346**, 1004–1006 (2014).
2. Acemel, R. D. *et al.* A single three-dimensional chromatin compartment in amphioxus indicates a stepwise evolution of vertebrate Hox bimodal regulation. *Nat. Genet.* **48**, 336–341 (2016).